# Class conditional conformal prediction for multiple inputs by p-value aggregation

**Jean-Baptiste Fermanian**
IMAG, IROKO, Univ Montpellier, Inria, CNRS,
Montpellier, France
`jean-baptiste.fermanian@inria.fr`

**Mohamed Hebiri**
LAMA, Université Gustave Eiffel,
Paris, France
`Mohamed.Hebiri@univ-eiffel.fr`

**Joseph Salmon**
IMAG, IROKO, Univ Montpellier, Inria, CNRS,
Montpellier, France
`joseph.salmon@inria.fr`

## Abstract

Conformal prediction methods are statistical tools designed to quantify uncertainty and generate predictive sets with guaranteed coverage probabilities. This work introduces an innovative refinement to these methods for classification tasks, specifically tailored for scenarios where multiple observations (multi-inputs) of a single instance are available at prediction time. Our approach is particularly motivated by applications in citizen science, where multiple images of the same plant or animal are captured by individuals. Our method integrates the information from each observation into conformal prediction, enabling a reduction in the size of the predicted label set while preserving the required class-conditional coverage guarantee. The approach is based on the aggregation of conformal p-values computed from each observation of a multi-input. By exploiting the exact distribution of these p-values, we propose a general aggregation framework using an abstract scoring function, encompassing many classical statistical tools. Knowledge of this distribution also enables refined versions of standard strategies, such as majority voting. We evaluate our method on simulated and real data, with a particular focus on Pl@ntNet, a prominent citizen science platform that facilitates the collection and identification of plant species through user-submitted images.

## 1 Introduction

As modern algorithms and the data they process become increasingly complex, the importance of quantifying their uncertainty has grown significantly. One systematic method to meet this requirement is *(split) conformal prediction*, which leverages a scoring function and labeled calibration data to generate a set of potential outputs for a new instance (Vovk et al., 2005). For the users, the size of this prediction set is intended to reflect the algorithm's uncertainty. This approach requires the assumption that the calibration data and new observations are exchangeable, (*i.e.,* invariant by permutations) to get a prescribed coverage. In classification tasks, the conformal set usually comprises a subset of labels that, with a high degree of prescribed confidence, includes the true label of the new observation (see Angelopoulos et al., 2023; Fontana et al., 2023 for overviews of conformal prediction and Chzhen et al., 2021 for set valued classification).

In this work, we focus on a setting where each input to be classified (at testing) consists of multiple observations, each of which can be independently processed prior to aggregation. This methodology is commonly employed in machine learning-driven citizen science applications, such as Pl@ntNet

(Joly et al., 2014), where users are prompted to submit one or more images of a plant for identification. Similar multi-input samples are also prevalent in iNaturalist (Van Horn et al., 2018) or in eBird (Sullivan et al., 2009), the latter managing multi-input sounds of birds, rather than images. Although multiple images may be submitted, algorithms often process them independently and only aggregate the resulting (softmax) predictions. A naive conformal prediction step based on such aggregated scores severely disrupts the exchangeability. This results in excessively large prediction sets, rendering them impractical (see Appendix B).

Simultaneously, we aim at class conditional coverage calling which would need a sufficiently large number of calibration examples for each class to reliably estimate how aggregation behaves. In datasets with a highly imbalanced class distribution, such as those commonly found in citizen science datasets with many underrepresented classes (see, for example, Garcin et al., 2021), achieving this requirement is not feasible.

To overcome this limitation, we propose to perform conformal prediction at the level of individual observations, and to aggregate the resulting conformal prediction sets. We frame this problem as one of p-value aggregation, where the p-values are the conformal p-values computed for each individual observation. Using recent results on the distribution of these p-values, we develop a general framework for aggregating them using score functions, which encompasses classical techniques such as p-value correction, majority voting, or template matching.

**Setting.** We consider the scenario of multi-class classification with $K$ classes. We have at disposal an already trained classifier $\mathcal{A} : \mathcal{X} \to [0, 1]^K$, a *calibration set* of $n$ labeled points $\{(X_i, Y_i)\}_{i=1}^n$, and multiple observations $\{X_{n+j}\}_{j=1}^m$ of the *same* item from the (unknown) class $Y_{n+1}$. Our objective is to build a set $\mathcal{C}_\alpha$ depending on the classifier, the calibration set, and the new points satisfying for a fixed $\alpha \in (0, 1)$ and for all $y \in [\![K]\!]$:

$$\mathbb{P}\Big[y \in \mathcal{C}_\alpha\Big(\mathcal{A}, \{(X_i, Y_i)\}_{i=1}^n, \{X_{n+j}\}_{j=1}^m\Big)\Big|Y_{n+1} = y\Big] \geq 1 - \alpha \ . \tag{1}$$

To get an informative set, $\mathcal{C}_\alpha$ is intended to be as small as possible.

**Notation.** For two integers $n \leq m$, we denote $[\![n : m]\!] := \{n, \ldots, m\}$ and $[\![n]\!] := [\![1 : n]\!]$. We set $s : \mathcal{X} \times [\![K]\!] \to \mathbb{R}$ to be a score function and $S_i = s(X_i, Y_i)$ for $i \in [\![n]\!]$ are the calibration scores. For a label $y \in [\![K]\!]$, we denote $n_y = |\{i \in [\![n]\!] : Y_i = y\}|$ the number of calibration points of class $y$ (supposed fixed in most of our results) and $S_1^y, \ldots, S_{n_y}^y$ the scores of these points. We denote $S_{n+i}(y) := s(X_{n+i}, y)$ the score of a label $y$ at the test point $X_{n+i}$ for $i \in [\![m]\!]$.

**Contributions.** We develop a framework for constructing a conformal prediction set that is class conditionally valid (1), by aggregating multiple observations from the same class. This approach can be viewed as stemming from tests that assess the exchangeability between the observations and the points within each class. It has been made possible through the adaptation of recent findings on the joint distribution of conformal p-values, as discussed (Gazin et al., 2023). Our contributions are summarized below:

**1.** The prediction set is built from class conditional conformal p-values for which we give the exact distribution. Notably, our construction explicitly handles *exchangeability* and, in particular, *ties* between the conformal scores—a frequent occurrence with softmax outputs from neural networks, yet often overlooked in the literature. This probabilistic result serves as a key ingredient in the contributions that follow (Section 3).

**2.** We relate our method to the prediction set aggregation framework of Gasparin and Ramdas (2024), and propose an enhancement tailored to the setting where the aggregated sets are class-conditional conformal sets constructed from a shared calibration set (Section 4).

**3.** We propose a new efficient way to aggregate class-conditional conformal p-values by constructing a region of reject based on the construction of a score $v$ on the p-values. (Section 5).

**4.** We test our methods on a classification on a synthetic dataset of mixture of Gaussian distributions and on `LifeCLEF Plant Identification Task 2015` (Goëau et al., 2015). In all the experiments, the classifiers are neural networks with softmax output (Section 6).

## 2 Background and related works

### 2.1 Overview on classical conformal prediction

Conformal Prediction (CP) is a framework introduced by Vovk et al. (2005) to construct distribution-free prediction sets quantifying the uncertainty in the predictions of an algorithm. In this context, $\alpha \in (0, 1)$ represents the significance level, and $1 - \alpha$ denotes the targeted coverage probability. Formally, given a calibration set $\{(X_i, Y_i)\}_{i \in [\![n]\!]}$ of points in $\mathcal{X} \times [\![K]\!]$ and a trained classifier, a CP method constructs for a new point $X_{n+1} \in \mathcal{X}$ a *marginally valid* set $\mathcal{C}_\alpha$, *i.e.,* a set containing the unobserved outcome $Y_{n+1} \in [\![K]\!]$ with high probability:

$$\mathbb{P}[Y_{n+1} \in \mathcal{C}_\alpha(X_{n+1})] \geq 1 - \alpha \ . \tag{2}$$

The idea behind CP methods is to first build non-conformity scores $S_i := s(X_i, Y_i)$ on the calibration set, for a given score function $s : \mathcal{X} \times [\![K]\!] \to \mathbb{R}$ intended to quantify the error of the algorithm at a given point. Many score functions have been considered to catch different type of information, see *e.g.,* Angelopoulos et al. (2023) for a review. A common choice in classification is to use (one minus) the estimated probability of the class (*e.g.,* the softmax output of neural network). In this paper, we will focus on the case where an already trained algorithm is provided, which corresponds to the case of split CP. Then, for a score function $s$, the prediction set

$$\mathcal{C}_\alpha(x) := \left\{ y \in [\![K]\!] \ : \ s(x, y) \leq S_{(\lceil (1-\alpha)(n+1) \rceil)} \right\} \ , \tag{3}$$

is marginally valid (as defined in Eq. (2)) where $S_{(k)}$ is the $k$-th smallest score of $S_1, \ldots, S_n, \infty$ (Vovk et al., 2005).

In certain applications, it is desirable to construct prediction sets that satisfy a *class conditional* coverage guarantee; that is, for every class $y$, the following condition holds:

$$\mathbb{P}[y \in \mathcal{C}_\alpha(X_{n+1}) \mid Y_{n+1} = y] \geq 1 - \alpha \ . \tag{4}$$

While this condition implies marginal coverage, the converse does not hold. In scenarios with highly imbalanced or heterogeneous classes, marginal split CP can lead to prediction sets with significant variability in class-wise coverage (Sadinle et al., 2019; Ding et al., 2023). To enforce conditional coverage as specified in (4), a natural approach is to apply split CP separately for each class (Vovk et al., 2005). The resulting prediction set is given by

$$\mathcal{C}_\alpha^{cd}(x) = \left\{ y : s(x, y) \leq S_{(\lceil (1-\alpha)(n_y+1) \rceil)}^y \right\} \ , \tag{5}$$

which is conditionally valid, with $\{S_i^y\}_{i \in [\![n_y]\!]}$ being the scores of the $n_y$ points with label $y$ in the calibration set. A limit of this approach is that it requires to have at least $\frac{1}{\alpha} - 1$ points for each class (Ding et al., 2023). Otherwise, such a class would always be predicted which can lead to uninformative prediction set.

### 2.2 P-value point of view of conformal prediction

The conformal set (3) can be interpreted as the outcome of a family of two-sample tests, each assessing—for every $y \in [\![K]\!]$—whether $s(X_{n+1}, y)$ is exchangeable with the scores of the calibration. The predicted classes $y \in \mathcal{C}_\alpha$ are those not rejected by a Wilcoxon-Mann-Whitney test (between a sample of size 1 and of size $n$, Wilcoxon, 1945; Mann and Whitney, 1947). One can associate a so-called split conformal p-value (Vovk et al., 2005) to split CP, defined as

$$p_{n+1}^{mrg}(y) = \frac{1}{n+1} \left[ \sum_{i=1}^n \mathbf{1}\{s(X_{n+1}, y) \leq S_i\} + 1 \right] \ , \tag{6}$$

such that the condition $s(X_{n+1}, y) \leq S_{(\lceil (1-\alpha)(n+1) \rceil)}$ in Eq. (3) is equivalent to $p_{n+1}(y) \geq \alpha$. For the true label $Y_{n+1}$, assuming the scores have *no ties*, the p-value of the true label $p(Y_{n+1})$ follows a uniform distribution over $\{i/(n+1) : i \in [\![n+1]\!]\}$. This distribution is explicit and independent of the underlying data distribution, which allows for effective identification of unlikely candidate labels. The construction of this distribution-free quantity forms the basis of conformal prediction (see Barber and Tibshirani, 2025 for a recent p-value point of view).

Similarly, for class-conditional prediction sets (5), a class-conditional p-value can be defined as $p_{n+1}^{cd}(y) = \frac{1}{n_y+1}\left[\sum_{i=1}^{n_y} \mathbf{1}\{s(X_{n+1}, y) \leq S_i^y\} + 1\right]$. As in the marginal case, this set can be interpreted as arising from a test of exchangeability between the test score and the calibration scores for the corresponding class.

## 2.3 Joint control of the p-values.

In our setting, where multiple images $X_{n+1}, \ldots, X_{n+m}$ with the same label $Y_{n+1}$ are observed, our approach relies on constructing vectors of conformal p-values $\mathbf{p}(y) = (p_1(y), \ldots, p_m(y))$, where $p_j(y)$ is the class-conditional p-value associated with $X_{n+j}$ for class $y$. One question is to find a set $E \subset [0, 1]^m$ that contains the vector $\mathbf{p}(Y_{n+1})$ with high probability:

$$\mathbb{P}[\mathbf{p}(Y_{n+1}) \in E] \geq 1 - \alpha \ . \tag{7}$$

For $m = 1$, conformal prediction chooses $E = [\alpha, 1]$, but, for $m$ larger, a wide range of choices become possible. This question has been extensively studied across various contexts and for multiple purposes. In the realm of multiple testing, numerous methods have been developed to control the false discovery proportion (FDP), often leading to criteria interpretable as statistical envelope: For instance Bonferroni (1936); Rüger (1978); Rüschendorf (1982); Vovk et al. (2022) address scenarios involving arbitrarily dependent p-values.; Fisher (1925); Pearson (1934); Simes (1986); Benjamini and Hochberg (1995) focus on settings with independent p-values. Additionally, Sarkar (1998); Benjamini and Yekutieli (2001) provide results under positive dependence assumptions while the exchangeable case has been recently considered by Gasparin et al. (2025). Blanchard et al. (2020) (see also Li et al., 2024) introduce the notion of *template*, which most closely aligns with the envelope perspective we adopt. In the framework of conformal prediction, controls of the form (7) have been employed across various applications, including outlier detection using conformal scores (Bates et al., 2023; Gazin et al., 2023), batch prediction (Gazin et al., 2024; Lee et al., 2024), aggregation of prediction intervals (Gasparin and Ramdas, 2024), and conformal prediction for ranking (Fermanian et al., 2025). All these methods can be summarized as constructing a score function $v$ of the p-values such that $\mathbf{p} \in E \iff v(\mathbf{p}) \geq q_E$, where $q_E$ is a threshold specific to each method. Table 1 provides an overview of these methods, the associated score functions $v$, and a concise summary of the required dependencies among the p-values to ensure the guarantee (7).

| | Set $E \subset [0,1]^m$ | $v(p)$ | Dependence |
|---|---|---|---|
| Bonferroni | $\{\mathbf{p} : \forall j, \ p_j \geq \alpha/m\}$ | $\min p_j$ | – |
| Simes | $\{\mathbf{p} : \forall j, \ p_{(j)} \geq j\alpha/m\}$ | $\min_j \ mp_{(j)}/j$ | PRDS |
| Template | $\{\mathbf{p} : \forall j, \ p_{(j)} \geq t_j(\lambda_\alpha)\}$ | $\min_j t_j^{-1}(p_{(j)})$ | – |
| Majority vote | $\{\mathbf{p} : \sum_{j=1}^m \mathbf{1}_{\{p_j \geq \alpha/2\}} \geq m/2\}$ | $\widehat{F}_m^{-1}(1/2)$ | – |
| | $\{\mathbf{p} : \forall k, \ \sum_{j=1}^k \mathbf{1}_{\{p_j \geq \alpha/2\}} \geq k/2\}$ | $\min_k \widehat{F}_k^{-1}(1/2)$ | Exchangeability |

Table 1: Examples of sets satisfying (7) and the associated score function. $\widehat{F}_k$ denotes the empirical CDF of $(p_1, \ldots, p_k)$. Template method refers to Blanchard et al. (2020) where $t_j$ denotes a family of functions and $\lambda_\alpha > 0$ is a parameter. Majority vote refers to the methods analyzed by Gasparin and Ramdas (2024). PRDS denotes Positive Regression Dependence on a Subset (Benjamini and Yekutieli, 2001).

## 3 Joint distribution of the conditional p-values

The present section introduces fundamental mathematical tools that will be required to derive conformal predictors (based on aggregation or on p-values) with the right validity in Section 4 and Section 5. We first present some key lemma about a uniform distribution over the ranks. We then link this distribution to the joint distribution of conditional p-values associated to each observation $X_{n+i}$. Finally, we derive the distribution of some key statistics.

**Key lemma.** Let $A_{n,m}$ for $n, m \in \mathbb{N}^*$, be the set of ordered discrete p-values:

$$A_{n,m} := \left\{ a \in \left\{ \frac{i}{n} : 0 \leq i \leq n \right\}^m : a_1 \leq \ldots \leq a_m \right\} \ . \tag{8}$$

**Lemma 3.1.** *Let $n, m \in \mathbb{N}^*$ and $P \sim \mathcal{U}(A_{n,m})$, where $\mathcal{U}$ denotes the uniform distribution. Then*

- $n \cdot P(j) \sim \texttt{BetaBin}(n, j, m - j + 1)$ *for* $j \in [\![m]\!]$.
- $\sum_{j=1}^{m} \mathbf{1}_{\{n \cdot P(j) \geq \ell\}} \sim \texttt{BetaBin}(m, n + 1 - \ell, \ell)$ *for* $\ell \in [\![0 : n]\!]$.

*Remark* 3.2. The beta-binomial distribution $\texttt{BetaBin}(m, a, b)$ is the distribution of a binomial random variable $\mathcal{B}(m, Q)$ where $Q$ is drawn independently as a beta distribution $B(a, b)$. If $a$ and $b$ are positive integers, then for $k \in [\![0 : m]\!]$,

$$\mathbb{P}\Big[\texttt{BetaBin}(m, a, b) = k\Big] = \frac{\binom{k+a-1}{k}\binom{m-k+b-1}{m-k}}{\binom{m-1+a+b}{m}} \ .$$

In this case, the distribution $\texttt{BetaBin}(m, a, b)$ is a negative hypergeometric distribution which can be easily simulated using Polya's urns. For $a = 0$ (resp. $b = 0$), the distribution is defined as a Dirac in 0 (resp. in $m$).

**Joint distribution of the conditional p-values.** We focus here on the joint distribution of the ordered vector of p-values $\mathbf{p}_\uparrow(y) = (p_{(1)}(y), \ldots, p_{(m)}(y))$, where for $j \in [\![m]\!]$ and $y \in [\![K]\!]$

$$p_j(y) = \frac{1}{n_y} \sum_{i=1}^{n_y} \mathbf{1}\{s(X_{n+j}, y) \leq S_i^y\} \ . \tag{9}$$

The guarantees we obtain on this vector are under the condition that the scores of the new points evaluated in the true label are exchangeable with the scores of the same class:

**Assumption 3.3.** *For all $y \in [\![K]\!]$ and conditionally to $Y_{n+1} = y$, the vector of scores $\left(S_1^y, \ldots, S_{n_y}^y, S_{n+1}(y), \ldots, S_{n+m}(y)\right)$ is exchangeable.*

This hypothesis is fundamental to our work. It assumes that the scores of the observed images behave similarly to the calibration scores of the same class. This assumption is more discussed in Section E. *Remark* 3.4. The number of multi-inputs $m$ is formally assumed to be fixed throughout the paper. However, the method remains directly applicable when $m$ varies. In that case, if Assumption 3.3 holds conditionally on $m$, then our results remain valid conditionally on $m$.

The distribution of the unordered vector of $p$-values has been deeply studied by Gazin et al. (2023); Gazin (2024) in the general case where the p-values are the marginal ones, the test points do not all share the same class, and the scores have no ties. The following proposition can be seen as an extension of these results conditionally to the class, which includes the possibility of *ties* between scores. To this purpose, we introduce a randomized version $\mathbf{p}_\uparrow^{\texttt{rd}}$ of the vector of conformal p-values $\mathbf{p}$.

**Theorem 3.5.** *Under Assumption 3.3, for $y \in [\![K]\!]$ and $j \in [\![m]\!]$, let us introduce*

$$p_j^{rd}(y) = \frac{1}{n_y} \sum_{i=1}^{n_y} \Big[\mathbf{1}\{S_{n+j}(y) < S_i^y\} + \mathbf{1}\{S_{n+j}(y) = S_i^y\}\mathbf{1}\{U_j \leq U_i^y\}\Big] \ , \tag{10}$$

*where $(U_i)_{i \in [\![m]\!]}$ and $(U_i^y)_{i \in [\![n_y]\!]}$ for $y \in [\![K]\!]$ are i.i.d. uniform random variables on $[0, 1]$. Then the ordered vector of randomized p-values $\mathbf{p}_\uparrow^{rd}(y) = \left(p_{(1)}^{rd}(y), \ldots, p_{(m)}^{rd}(y)\right)$, conditionally to $Y_{n+1} = y$, is uniformly drawn from $A_{n_y,m}$:*

$$\mathbf{p}_\uparrow^{rd}(Y_{n+1}) \,|\, (Y_{n+1} = y) \sim \mathcal{U}\big(A_{n_y,m}\big) \ . \tag{11}$$

*Moreover, for $j \in [\![m]\!]$, we have $n_y \cdot p_{(j)}^{rd}(Y_{n+1}) \,|\, (Y_{n+1} = y) \sim \texttt{BetaBin}(n_y, j, m - j + 1)$.*

The proof is given in Appendix A.3. As a minor remark, notice that when scores have no ties, the original $\mathbf{p}$ is preserved. More important, the advantage of the above result is that the knowledge of the exact distribution of the conformal p-values will allow us to estimate more precisely the quantiles of statistics constructed from this vector. As a consequence, the resulting prediction sets will be more informative (smaller sets) than those based on bounds for these quantiles.

*Remark* 3.6. The definition (10) of the conformal p-values differs a bit from the usual definition (different normalization and no additional $+1$, can be compared with (6)) but provides an equivalent tool. With this definition, the ordered vector of p-values has a symmetric distribution around the diagonal: $1 - (p_{(m)}(Y_{n+1}), \ldots, p_{(1)}(Y_{n+1}))$ has the same distribution as $\mathbf{p}_\uparrow^{\texttt{rd}}(Y_{n+1})$.

**Other important law derivations.** Combining this first result with Lemma 3.1, we can derive the distribution of critical quantities such as the empirical coverage or the marginal distribution of the ordered vector.

**Corollary 3.7.** *Under Assumption 3.3, let $B_y = \sum_{j=1}^{m} \mathbf{1}\{y \in \mathcal{C}_{\alpha}^{cd}(X_{n+j})\}$ be the number of class conditional sets containing the label $y$. Then for $y \in [\![K]\!]$ and $\ell \in [\![m]\!]$ :*

$$\mathbb{P}\big[B_{Y_{n+1}} = \ell | Y_{n+1} = y\big] \geq \mathbb{P}[\beta = \ell] , \tag{12}$$

*where $\beta$ follows the distribution* `BetaBin`$(m, (n_y + 1) - k_{y,\alpha}, k_{y,\alpha})$ *with* $k_{y,\alpha} = \lfloor (n_y + 1)\alpha \rfloor$. *Moreover, if the scores have no ties, Eq. (12) is an equality.*

The distribution of empirical coverage for conformal prediction sets has been first studied by Vovk (2012) and Angelopoulos et al. (2023). For the exchangeable case with no ties, we will refer to Marques F. (2025) who obtains a similar result. This result will be used to enhance the aggregation of conformal prediction sets by majority voting in our setting.

## 4 Refinement of majority voting for multiple inputs

Thanks to these probabilistic results, we are in position to build new conformal predictors tailored for scenarios where multiple predictive observations of a single instance are available. As this problem can be seen as the aggregation of the prediction set associated to each instance, we focus in this section to the adaptation to our setting of the general majority voting methods proposed by Gasparin and Ramdas (2024). They propose to keep the labels appearing in at least one half of the sets, in our setting it consists to consider the *majority voting* set $\mathcal{C}_{\alpha,m}^{M} = \{y : \sum_{j=1}^{m} \mathbf{1}\{y \in \mathcal{C}_{\alpha}^{cd}(X_{n+j})\} \geq m/2\}$. These authors also propose *exchangeable majority* voting sets defined as $\mathcal{C}^{Me} = \cap_{k=1}^{m} \mathcal{C}_{\alpha,k}^{M}$ usable in our setting as the sets $\mathcal{C}_{\alpha}^{cd}(X_{n+j})$ are exchangeable. Both methods achieve a class-conditional coverage of $1 - 2\alpha$, replacing $\alpha$ by $\alpha/2$ leads to the correct coverage (see Appendix C).

If $\mathcal{C}_j = \mathcal{C}_{\alpha}^{cd}(X_{n+j})$, then thanks to Proposition 3.7, the distribution of the number of sets containing the true label $Y_{n+1}$ is known. The threshold $m/2$ chosen in the majority vote method can then be improved: we replace it by the quantile of a Beta-Binomial distribution.

**Proposition 4.1.** *Let $\alpha \in (0, 1)$, assume Assumption 3.3, let*

$$\mathcal{C}_{\alpha}^{BB} := \left\{y : \sum_{i=1}^{m} \mathbf{1}\{y \in \mathcal{C}_{\alpha}^{cd}(X_{n+i})\} \geq q_{\alpha}^{BB}(y)\right\} , \tag{13}$$

*where $q_{\alpha}^{BB}(y)$ is the $\alpha$-quantile of the distribution* `BetaBin`$(m, \lceil (n_y + 1)(1 - \alpha) \rceil, \lfloor (n_y + 1)\alpha \rfloor)$. *Then, for all $y \in [\![K]\!]$:*

$$\mathbb{P}\big[Y_{n+1} \in \mathcal{C}_{\alpha}^{BB} \,|\, Y_{n+1} = y\big] \geq 1 - \alpha .$$

For each class, the set $\mathcal{C}_{\alpha}^{BB}$ requires computing or storing quantiles of a Beta-Binomial distributions, which depends on the number of calibration points belonging to that class. To avoid performing this computation for each class individually, for example if the number of classes is too large, one can approximate the Beta-Binomial distribution by a Binomial distribution (see Lemma F.1). This approximation induces a loss in coverage as presented in the following proposition.

**Proposition 4.2.** *Let $\alpha \in (0, 1)$, assume Assumption 3.3 holds, let*

$$\mathcal{C}_{\alpha}^{Bin} := \left\{y : \sum_{j=1}^{m} \mathbf{1}\{y \in \mathcal{C}_{\alpha}^{cd}(X_{n+j})\} \geq q_{\alpha}^{Bin}\right\} , \tag{14}$$

*where $q_{\alpha}^{Bin}$ is the $\alpha$-quantile of the binomial distribution $\mathcal{B}(m, 1 - \alpha)$. Then for all $y \in [\![K]\!]$:*

$$\mathbb{P}\big[Y_{n+1} \in \mathcal{C}_{\alpha}^{Bin} | Y_{n+1} = y, n_y\big] \geq 1 - \alpha - \frac{(m-1)\min(1, \alpha m)}{n_y + 1} , \quad a.s. \tag{15}$$

*Moreover, if the labels $Y_1, \ldots, Y_{n+1}$ are i.i.d., then*

$$\mathbb{P}\big[Y_{n+1} \in \mathcal{C}_{\alpha}^{Bin}\big] \geq 1 - \alpha - \frac{K(m-1)\min(1, \alpha m)}{n+1} . \tag{16}$$

The loss of coverage remains negligible when the number of repetitions is small ($m \ll n/K$ for the marginal one). In the setting of Pl@ntNet data, the number of repetitions is effectively low, but some classes have also a small numbers of examples in the calibration set which makes this approximation only usable for large classes or when only a marginal coverage is targeted.

*Remark* 4.3. The possibility to use a binomial distribution quantile has been already proposed by Gasparin and Ramdas (2024) to merge independent prediction sets. The assumption of independence is a bit restrictive and this result proves that it can effectively be used for prediction sets correlated by the calibration (so not independent) for small enough number of sets $m$.

## 5 p-value aggregation methods

Voting methods presented above aggregate the conformal sets constructed for each observation point by only considering the number of sets that contain a given label. Although this procedure can be interpreted as an aggregation of the conformal p-values and as forming an envelope around them (see Figure 1), we observe, however, that some information is lost in the process. In this section, we exploit the structure of the ordered vector of p-values to aggregate the observations. These methods decisively outperform the previously discussed voting strategies.

### 5.1 General method

The method consists in building a (non-)conformity score function $v : [0,1]^m \times [\![K]\!] \to \mathbb{R}$ directly on the p-values vector. Then, exploiting the result in Theorem 3.5, we calibrate this new score by simulating samples $P_1^y, \ldots, P_T^y$ of the distribution $\mathcal{U}(A_{n_y,m})$ for each class, using for example Algorithm 1. Here, $T$ can be seen as a budget for approximating the score's distribution by Monte Carlo sampling. The method is summarized in the following result which gives a guarantee on the conditional coverage.

**Proposition 5.1.** *Let $v : [0,1]^m \times [\![K]\!] \to \mathbb{R}$ and $T \in \mathbb{N}$, for $y \in [\![K]\!]$ let $P_t^y \overset{i.i.d.}{\sim} \mathcal{U}(A_{n_y,m})$ and $V_t^y := v(P_t^y, y)$ for $t \in [\![T]\!]$. Then, for $\alpha \in (0,1)$, if Assumption 3.3 is satisfied, the set*

$$\mathcal{C}_\alpha^v = \left\{ y : v(\mathbf{p}_\uparrow^{rd}(y), y) \geq V_{(\lfloor (T+1)\alpha \rfloor)}^y \right\} \tag{17}$$

*is conditionally valid: $\mathbb{P}[y \in \mathcal{C}_\alpha^v | Y_{n+1} = y] \geq 1 - \alpha$ for all $y \in [\![K]\!]$.*

This result is a direct application of conformal prediction using the score function $v$, and follows as a consequence of Theorem 3.5. While the choice of the score function $v$ is discussed below, it is worth noting that for a given $v$, we retain the low scores in order to remain within the standard framework of envelope methods: as shown in Table 1, the scores associated with envelopes tend to reject p-values corresponding to low scores.

---

**Algorithm 1** Simulation of $P \sim \mathcal{U}(A_{n,m})$

1: **Input:** $n, m$.
2: Draw $R_1, \ldots, R_m$ w./o. replacement from $[\![n+m]\!]$.
3: $P_i \leftarrow R_{(i)}$ for $1 \leq i \leq m$.
4: $P_i \leftarrow P_i - i$ for $1 \leq i \leq m$.
5: **Output:** $(P_1/n, \ldots, P_m/n)$.

---

*Remark* 5.2. Recall that the guarantees provided by conformal prediction in (2) and (4) hold with high probability, including with respect to the calibration set. Formally, the sample $(P_t^y)_{t,y}$ should be redrawn at each execution. However, it is also possible to fix $T$ sufficiently large and store the value $V_{(\lfloor (T+1)\alpha \rfloor)}^y$ for each class, interpreting it as a quantile estimate of the score distribution.

### 5.2 Construction of score functions $v$

We propose in the following different envelopes - associated to a score function $v : [0,1]^m \to \mathbb{R}$ on the p-values—that can either be used for the vector of p-values $\mathbf{p}$ or its randomized version $\mathbf{p}_\uparrow^{rd}$ as in the above result.

**Quantile envelope.** The first envelope that we consider relies on quantiles of the beta-binomial distribution. For a clear presentation, let us first remark that as presented in Theorem 3.5, the marginal distributions of the sorted vector of p-values $\mathbf{p}_\uparrow^{rd}(Y_{n+1})$ are known. Indeed, conditionally to $Y_{n+1} = y$ and $n_y$, the random variable $n_y \cdot p_j^{rd}(y)$ follows the distribution $\texttt{BetaBin}(n_y, j, n_y - j + 1)$.

To capture the geometry of the trajectory of p-values, we propose to choose an envelope of the trajectories constructed as the vector of quantiles of level $\lambda$ of each marginal (see Figure 1). Consider $\{F_j\}_j$ a general family of cdf and let us remark that

$$\left\{\mathbf{p} : \forall j : p_{(j)} \geq F_j^{(-1)}(\lambda)\right\} = \left\{\mathbf{p} : \min_j F_j(p_{(j)}) \geq \lambda\right\} .$$

The envelope of quantile of marginal associated to the *quantile* score $v_Q$ is defined as

$$v_Q(\mathbf{p}, y) := \min_{j \in [\![m]\!]} F_{j,n_y}(n_y \cdot p_{(j)}(y)), \quad \text{for } \mathbf{p} \in [0,1]^m, y \in [\![K]\!] , \tag{18}$$

where $F_{j,n}$ is the cdf of the distribution `BetaBin`$(n, j, m - j + 1)$. This kind of envelope, constructed as the true quantile of the marginal with a specific level, appears for instance in Genovese and Wasserman (2006); Blanchard et al. (2020) and in Fermanian et al. (2025) in an empirical version.

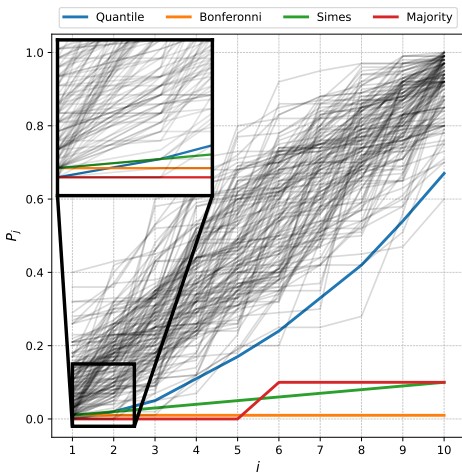

**$\ell^q$-Area envelope.** A simple idea to reject the classes whose p-values are too small could be supported by the fact that the area of the trajectory is too small. For $q > 0$, we define the *area* score: $v_{q-Area}(\mathbf{p}, y) := \|\mathbf{p}\|_q$, where $\|\cdot\|_q$ for $q \geq 1$ is the $\ell_q$-norm in $\mathbb{R}^m$. For $q = 1$, the procedure that consists in rejecting a trajectory or not can be viewed as a variant of the Wilcoxon-Mann-Whitney test (Wilcoxon, 1945; Mann and Whitney, 1947).

**$\ell^q$-envelope.** As asymptotically, the vector $\mathbf{p}_\uparrow^{\mathrm{rd}}(Y_{n+1})$ converges to the identity vector $\mathrm{Id} = (j/(m + 1))_{j=1}^m$, our last score only considers

Figure 1: Simulation of 150 samples of $P \sim \mathcal{U}(A_{n,m})$ for $m = 10$ and $n = 100$ (black lines). Quantile is the envelope associated to $v_Q$, and Majority to the majority vote.

the distance of p-value vector to this vector. In particular, $\mathbb{E}\left[p_{(j)}^{\mathrm{rd}}(Y_{n+1})\right] = \frac{j}{m+1}$ (*cf.* Theorem 3.5) which motivates the score $v_{\ell^q}$ defined as: $v_{\ell^q}(\mathbf{p}, y) := -\|\mathbf{p} - \mathrm{Id}\|_q$. The resulting set (17) will then keep p-values close to $\mathrm{Id}$. For $q = \infty$, bounds of the exact quantile and the asymptotic distribution are provided respectively by Gazin et al. (2023) and Gazin (2024).

# 6 Experiments

A key requirement of a conformal approach is its ability to accurately quantify uncertainty. For an efficient classifier, this should lead to small prediction sets; for a less accurate one, the conformal set should still contain the correct label, at the cost of a larger set. To this purpose, we compare the methods, by comparing the size of the prediction sets returned for a fixed coverage.

Multiple observations of the same instance are particularly valuable in challenging tasks where a single observation does not permit informative classification. As we will demonstrate, our aggregation techniques lead to a reduction in prediction set size in such cases resulting in a gain of interpretability. Throughout our experiments, we report the accuracy of the base classifiers. The score used for all the following experiments is the *softmax score* which is one minus the softmax output of the class. Additional results using the APS score (Romano et al., 2020) are presented in Section D.2.

The tested prediction sets are the *Majority* $\mathcal{C}^M$—Eq. (22), *Exchangeable Majority* $\mathcal{C}^{Me}$—Eq. (23), *BetaBinomial* $\mathcal{C}^{BB}$—Eq. (13), *Binomial* $\mathcal{C}^{Bin}$—Eq. (14), and the set $\mathcal{C}^v$—Eq. (17) with the score functions $v_Q$ (*Quantile*), $v_{1-Area}$ (*Wilcoxon*), $v_{2-Area}$ ($\ell_2$ *Area*) and $v_{\ell_2}$ ($\ell_2$).

Code for reproducing our experiments is available at https://github.com/jeanbaptistefermanian/Class_Conditional_CP_for_Multi_Inputs.

## 6.1 Synthetic data

We choose as distribution of the points, a mixture of Gaussian distributions with randomly drawn centers. Formally, $Y \sim \sum_{y=1}^K p(y)\delta_y$ and $X|(Y = y) \sim \mathcal{N}(\mu_y, \sigma^2 I_d)$, where $K = 10$, $d = 6$, the

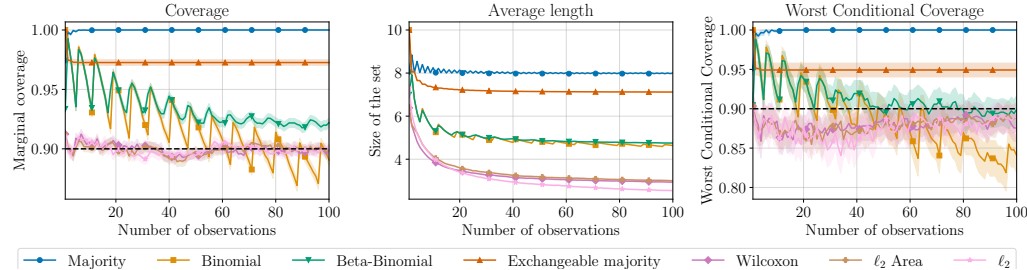

Figure 2: Synthetic data: marginal coverage, average length and minimum of the class conditional coverages in function of the number of observations for $\alpha = 0.1$, computed on 5000 repetitions. The confidence region is $\pm$ (empirical std)$/\sqrt{5000}$.

classes are a bit imbalanced $p(y) \propto (1 + 3(y-1)/K)^{-1}$ and $\sigma^2 = 3$. The centers of the classes $\mu_y$ are drawn independently as $\mathcal{N}(0, I_d)$. The strength of the noise makes this problem difficult with only one observation as the distance between two centers clusters is relatively close to the distance of a point to its cluster's center : $\mathbb{E}[\|X - \mu_Y\|] \simeq \mathbb{E}[\|\mu_y - \mu_z\|]$. The model is a ReLu neural network with three hidden layers of width 15. It is trained with 2000 observations and achieves a top 1 accuracy of 0.49. We apply our conformal procedure with a calibration set of size $n = 1000$ and repeat 5000 times our procedure for a number of observations $m \in \{1, \ldots, 100\}$.

We observe in Figure 2 that the methods based on the p-values perform best as compared to majority vote methods. For instance, for $m = 20$ and $\ell_2$-envelope method, we observe a significant improvement with a set of average size of 3.6 as compared to an original value of 6.6. They provide the smallest sets while maintaining a marginal coverage above $1 - \alpha$, and exhibit conditional coverage close to $1 - \alpha$. The majority vote methods appear to be overly conservative. The proposed refinement of the majority vote improves on the two original versions. As announced in Proposition 4.2, for small $m$, the binomial quantile approximates the beta-binomial one well. The marginal coverage is not significantly affected as long as $m/n$ remains small. However, in the third figure, the conditional coverage for some classes is impacted by this approximation when $m$ is large as the numbers of points of some classes in the calibration set are smaller than $m$. The over-coverage observed with the Beta-Binomial approach arises from the potential non-existence of a quantile at the exact level $1 - \alpha$. This issue can be addressed by introducing additional randomization.

## 6.2 The LifeCLEF Plant Identification Task 2015

We apply our method to `LifeCLEF Plant Identification Task 2015`[1] (Goëau et al., 2015) dataset, a dataset of 113,205 images of plants spread between 1K species (classes), with high heterogeneity. We split equally between train, calibration and test sets the original full dataset thanks to a stratified approach to maintain the original classes distribution. The classifier considered is ResNet50, trained on Pl@ntNet-300K Garcin et al. (2021)[2], for which we have fine tuned the last layer on our training set. It achieves a top-1 accuracy of $43\%$ (and top-5 of $63\%$, see Section D.1 for others

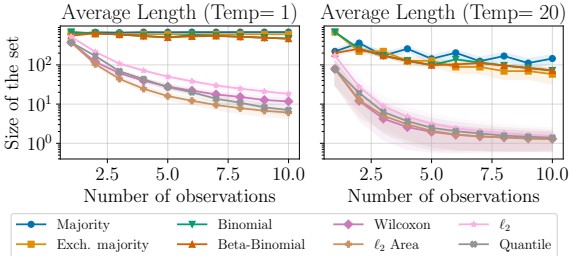

Figure 3: `LifeCLEF Plant Identification Task 2015`: average length in log-scale for $\alpha = 0.1$ and two different choice of temperature in function of the number of observations.

top-k accuracy). As a pre-processing step, we restrict ourselves to the classes (species) for which we have at least 20 examples in the calibration set, hence keeping $K = 688$ classes. This is needed for having informative class conditional CP set. We group randomly images of the same classes to simulate multi-view observations of the same plant with a number of repetitions of $m \in \{1, \ldots, 10\}$.

---

[1]https://www.imageclef.org/lifeclef/2015/plant

[2]see code and models at https://github.com/plantnet/PlantNet-300K/

In Figure 3, the prediction set lengths of various methods for two temperature settings (Temp=1 and Temp=20), all methods achieve a valid coverage of at least $0.9$, with detailed results deferred to Section D.2. The temperature refers to the standard re-scaling of the output of the network before the application of the softmax function. Our interpretation is that, due to the numerical approximation of the softmax function, a low temperature can lead to *ties* between scores (many are then numerically $0$ or $1$), which may hinder the performance of conformal procedures (see, *e.g.,* Dabah and Tirer, 2024 for a discussion on the relationship between temperature scaling and conformal prediction). Specifically, we observe that when using the raw network outputs without temperature adjustment (Temp=1), the majority vote struggles and tends to predict nearly all classes. In contrast, methods based on conformal p-values consistently produce narrower prediction sets after just a few observations and are less sensitive to this parameter. Our refinements to the majority vote method do lead to improvements over the original approach, but they are still outperformed by the p-value aggregation methods.

## 7    Conclusion and limitations

We introduced a new class-conformal prediction framework for the problem of classification when multiple observations of the same instance at prediction time are available. Assuming the scores of the new points evaluated in the true label are exchangeable with those of the same class, we developed an analysis of the distribution of the vector of the underlying ordered conformal p-values. We built two novel aggregation strategies of these p-values and showed how effective is the method that captures the best the structure of the conformal p-values.

Exchangeability of scores vector seems to be the bedrock of our strategy and more generally of conformal prediction (see Section E). We ensured this condition by sampling the multi-inputs randomly from the same class. It would be interesting to investigate the more realistic case of dependent multi-inputs, for which our method constitutes a foundational step.

## Acknowledgments and Disclosure of Funding

The authors acknowledge Ulysse Gazin, Pierre Humbert and Etienne Roquain for helpful discussion. This work was funded by the French National Research Agency (ANR) through the grant Chaire IA CaMeLOt (ANR-20-CHIA-0001-01).

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

# A  Proofs

## A.1  Preliminary results

This first result is derived from Theorem A.1 of Gazin et al. (2023).

**Corollary A.1.** *Let $S_1, \ldots, S_{n+m}$ exchangeable random variables with no ties ($S_i \neq S_j$ almost surely). For $j \in [\![m]\!]$, let*

$$p_j = \frac{1}{n} \sum_{i=1}^{n} \mathbf{1}\{S_{n+j} \geq S_i\}, \tag{19}$$

*and $\mathbf{p}_\uparrow = \left(p_{(1)}, \ldots, p_{(m)}\right)$ be the vector of ordered p-values. Then*

$$\mathbf{p}_\uparrow \sim \mathcal{U}(A_{n,m}),$$

*where $A_{n,m}$ is defined in (8).*

*Proof.* This result is a direct consequence of Theorem A.1 of Gazin et al. (2023). According to this work, let $M \in [\![0, m]\!]^n$ be the histogram of $\mathbf{p}$, *i.e.,* for $j \in [\![0, n]\!]$:

$$M_j = \left| \left\{ i : p_i = \frac{j}{n} \right\} \right|.$$

Then, $M \sim \mathcal{U}(H_{n,m})$ where $H_{n,m} = \{h \in [\![0, m]\!]^n : \sum_{i=1}^{n} h_i = m\}$ is the set of histogram (Gazin et al., 2023, Theorem A.1). We can now remark that $H_{n,m}$ is in bijection with the set of ordered trajectories $A_{n,m} = \left\{ a \in \left\{ \frac{i}{n} : 0 \leq i \leq n \right\}^m : a_1 \leq \ldots \leq a_m \right\}$. For example $\phi : H_{n,m} \to A_{n,m}$ defined for $h \in H_{n,m}$ by:

$$\phi(h) = \frac{1}{n} \left( \underbrace{0, \ldots, 0}_{h_0 \text{ times}}, \ldots, \underbrace{n, \ldots, n}_{h_n \text{ times}} \right),$$

is such a bijection. Then $\mathbf{p}_\uparrow = \phi(M)$ is uniformly drawn into $A_{n,m}$. $\qquad\square$

## A.2  Proof of Lemma 3.1

**Distribution of the marginal.** Let $P \sim \mathcal{U}(A_{n,m})$, $k \in [\![m]\!]$ and $j \in [\![0 : n]\!]$. The probability of the event $\{P(j) = k/n\}$ is the probability that the sub-trajectories $P_{1:(j-1)} = (P(1), \ldots, P(j-1))$ belong to $\{i/n, i \in [\![0 : k]\!]\}^{j-1}$ and the sub-trajectories $P_{(j+1):m} = (P(j+1), \ldots, P(m))$ belong to $\{i/n, i \in [\![k : n]\!]\}^{m-j}$. There are respectively $\binom{k+j-1}{j-1}$ and $\binom{n-k+m-j}{m-j}$ of such trajectories which are the numbers of non-decreasing sequences of $j - 1$ (resp. $m - j$) elements of $[\![0 : k]\!]$ (resp. $[\![k : n]\!]$). Then,

$$
\begin{aligned}
\mathbb{P}[P(j) = k/n] &= \frac{\#\{P \in A_{n,m} : nP_{1:(j-1)} \in [\![0 : k]\!]^{j-1}, nP(j) = k, nP_{(j+1):m} \in [\![k : n]\!]^{m-j}\}}{\#A_{n,m}} \\
&= \frac{\#A_{k,j-1} \#A_{n-k,m-j}}{\#A_{n,m}} \\
&= \frac{\binom{j+k-1}{k}\binom{n+m-j-k}{n-k}}{\binom{n+m}{n}} = \mathbb{P}\Big[\texttt{BetaBin}(n, j, m - j + 1) = k\Big].
\end{aligned}
$$

**Distribution of the empirical cdf.** The event $\left\{ \sum_{j=1}^{m} \mathbf{1}_{n \cdot P(j) \geq \ell} = k \right\}$ is satisfied if the first $m - k$ values of $P$ are strictly below $\ell/n$ and the $k$ last ones are above $\ell/n$. That means that the sub-trajectories $P_{1:(m-k)} = (P(1), \ldots, P(m-k))$ belong to $\{i/n, i \in [\![0 : \ell - 1]\!]\}^{m-k}$ and the sub-trajectories $P_{(m-k+1):m} = (P(m - k + 1), \ldots, P(m))$ belong to $\{i/n, i \in [\![\ell : n]\!]\}^{k}$. There are

respectively $\binom{\ell-1+m-k}{m-k}$ and $\binom{n-\ell+k}{k}$ of such trajectories . Then:

$$
\mathbb{P}\left[\sum_{j=1}^{m}\mathbf{1}_{n\cdot P(j)\geq\ell}=k\right] = \frac{\#\left\{P\in A_{n,m}: nP_{1:(m-k)}\in[\![0:\ell-1]\!]^{m-k}, nP_{(m-k+1):m}\in[\![\ell:n]\!]^{k}\right\}}{\#A_{n,m}}
$$

$$
= \frac{\#A_{\ell-1,m-k}\#A_{n-\ell,k}}{\#A_{n,m}}
$$

$$
= \frac{\binom{\ell-1+m-k}{m-k}\binom{n-\ell+k}{k}}{\binom{n+m}{m}} = \mathbb{P}\Big[\texttt{BetaBin}(m,n+1-\ell,\ell)=k\Big].
$$

$\square$

## A.3 Proofs of Theorem 3.5 and Corollary 3.7

In this section we prove Theorem 3.5 and Theorem 3.7 simultaneously. Let us first remark that if the scores have no ties, under the Assumption 3.3, all the results are obtained by applying Corollary A.1 and Lemma 3.1 conditionally to $(Y_{n+1}=y)$ to the scores $(S_1^y,\ldots,S_{n_y}^y,S_{n+1}(y),\ldots,S_{n+m}(y))$.

Let us now consider the case where there are potentially ties between the scores. The proof uses some common idea with the proof of Proposition 1 of Fermanian et al. (2025). Let $\delta$ be the minimum non null distance between the new scores of the true class and those evaluated in the calibration set:

$$
\delta := \min\Big\{|s-s'|, s\neq s' \in \{S_i\}_{i\in[\![n]\!]} \cup \{S_{n+j}(Y_{n+1})\}_{j\in[\![m]\!]}\Big\}.
$$

If all the scores are equal, $\delta$ can be set to any non null value (for example 1). We now define some proxy scores $\widetilde{S}$ by:

$$
\widetilde{S}_i^y := S_i^y + U_i^y\delta, \quad \text{for } y\in[\![K]\!], i\in[\![n_y]\!], \qquad \widetilde{S}_{n+j} := S_{n+j} + U_j\delta, \quad \text{for } j\in[\![m]\!], \quad (20)
$$

where $U_i^y, U_j$ for $y\in[\![K]\!]$, $i\in[\![n_y]\!]$ and $j\in[\![m]\!]$ are i.i.d. uniform random variables in $[0,1]$. The proxy scores $\widetilde{S}$ satisfy Assumption 3.3 and have no ties. We can then apply Corollary A.1 to the proxy scores $(\widetilde{S}_1^y,\ldots,\widetilde{S}_{n_y}^y,\widetilde{S}_{n+1}(y),\ldots,\widetilde{S}_{n+m}(y))$ conditionally to $(Y_{n+1}=y)$ to get (11). Indeed, the class conditional p-values $\widetilde{p}$ associated to the proxy scores $\widetilde{S}$ is equal to $p^{\texttt{rd}}$:

$$
\widetilde{p}_j(y) := \frac{1}{n_y}\sum_{i=1}^{n_y}\mathbf{1}\Big\{\widetilde{S}_{n+j}(y)\leq\widetilde{S}_i^y\Big\}
$$

$$
= \frac{1}{n_y}\sum_{i=1}^{n_y}\Big[\mathbf{1}\{S_{n+j}(y)<S_i^y\}+\mathbf{1}\{S_{n+j}(y)=S_i^y\}\mathbf{1}\{\delta U_j\leq\delta U_i^y\}\Big]
$$

$$
= \frac{1}{n_y}\sum_{i=1}^{n_y}\Big[\mathbf{1}\{S_{n+j}(y)<S_i^y\}+\mathbf{1}\{S_{n+j}(y)=S_i^y\}\mathbf{1}\{U_j\leq U_i^y\}\Big] = p_j^{\texttt{rd}}(y) .
$$

Then the distribution of the marginal $p_{(j)}^{\texttt{rd}}$ is a consequence of Lemma 3.1.

To get Corollary 3.7, we can remark that the proxy scores preserve the order as $S_{n+j}(y)<S_i^y \implies \widetilde{S}_{n+j}(y)<\widetilde{S}_i^y$. Thus for $j\in[\![m]\!]$ and $y\in[\![K]\!]$:

$$
y\in\mathcal{C}_\alpha^{cd}(X_{n+j}) \iff S_{n+j}(y)\leq S_{(\lfloor(n_y+1)(1-\alpha)\rfloor)}(y) \iff \sum_{i=1}^{n_y}\mathbf{1}\{S_{n+j}(y)\leq S_i^y\}\geq\lfloor(n_y+1)\alpha\rfloor
$$

$$
\iff \sum_{i=1}^{n_y}\mathbf{1}\{S_{n+j}(y)<S_i^y\}+\sum_{i=1}^{n_y}\mathbf{1}\{S_{n+j}(y)=S_i^y\}\geq\lfloor(n_y+1)\alpha\rfloor
$$

$$
\impliedby \sum_{i=1}^{n_y}\mathbf{1}\Big\{\widetilde{S}_{n+j}(y)<\widetilde{S}_i^y\Big\}\geq\lfloor(n_y+1)\alpha\rfloor
$$

$$
\iff \sum_{i=1}^{n_y}\mathbf{1}\Big\{\widetilde{S}_{n+j}(y)\leq\widetilde{S}_i^y\Big\}\geq\lfloor(n_y+1)\alpha\rfloor
$$

$$
\iff y\in\widetilde{\mathcal{C}}_\alpha^{cd}(X_{n+j}) ,
$$

where $\widetilde{\mathcal{C}}_\alpha^{cd}$ is the class conditional prediction set (5) constructed with the proxy scores $\widetilde{S}$. We have therefore shown that $\widetilde{\mathcal{C}}_\alpha^{cd}(X_{n+j}) \subset \mathcal{C}_\alpha^{cd}(X_{n+j})$, using that the randomization preserves the order and that the scores "tilde" have no ties. This implies that almost surely, the empirical coverage of the true scores is lower bounded by the empirical coverage of conformal sets constructed with the proxy scores, *i.e.,* :

$$\sum_{j=1}^m \mathbf{1}\{Y_{n+1} \in \mathcal{C}_\alpha^{cd}(X_{n+j})\} \geq \sum_{j=1}^m \mathbf{1}\left\{Y_{n+1} \in \widetilde{\mathcal{C}}_\alpha^{cd}(X_{n+j})\right\} .$$

We conclude again using Lemma 3.1.

### A.4 Proof of Proposition 4.1

The result is a direct consequence of Corollary 3.7. $\qquad\square$

### A.5 Proof of Proposition 4.2

Let us denote $B_y = \sum_{j=1}^m \mathbf{1}\{y \in \mathcal{C}_\alpha^{cd}(X_{n+j})\}$. First observe that:

$$\{Y_{n+1} \notin \mathcal{C}_\alpha^{Bin}\} = \{B_{Y_{n+1}} < Q_\alpha(\mathcal{B}(m, 1 - \alpha))\} \subset \{B_{Y_{n+1}} < Q_\alpha(\mathcal{B}(m, 1 - \alpha_{Y_{n+1}}))\},$$

where $\alpha_{Y_{n+1}} = \frac{\lfloor (n_{Y_{n+1}}+1)\alpha \rfloor}{n_{Y_{n+1}}+1} < \alpha$ and $Q_\alpha(\mathbb{Q})$ denotes the quantile $\alpha$ of the distribution $\mathbb{Q}$. Let $Z_y \sim \mathcal{B}(m, 1 - \alpha_y)$ and $\beta_y \sim \texttt{BetaBin}(m, n_y + 1 - k_y, k_y)$ where $k_y = \lfloor \alpha(n_y + 1) \rfloor$. Then, for $y \in [\![K]\!]$:

$$\begin{aligned}
\mathbb{P}\big[Y_{n+1} \notin \mathcal{C}_\alpha^{Bin}|Y_{n+1} = y\big] &\leq \mathbb{P}[B_y < Q_\alpha(\mathcal{B}(m, 1 - \alpha_y))|Y_{n+1} = y] \\
&\leq \mathbb{P}[\beta_y < Q_\alpha(\mathcal{B}(m, 1 - \alpha_y))],
\end{aligned}$$

using Corollary 3.7. It follows that:

$$\begin{aligned}
\mathbb{P}\big[Y_{n+1} \notin \mathcal{C}_\alpha^{Bin}|Y_{n+1} = y\big] \leq &\mathbb{P}[Z_y < Q_\alpha(\mathcal{B}(m, 1 - \alpha_y))] \\
&+ d_{TV}\Big(\mathcal{B}(m, 1 - \alpha_y), \texttt{BetaBin}(m, n_y + 1 - k_y, k_y)\Big) \\
\leq &\alpha + \frac{(m-1)\min(1, \alpha m)}{n_y + 1} , \qquad\qquad (21)
\end{aligned}$$

where we have used Corollary F.2 in the last inequality to bound the total variation distance between the Binomial and the Beta-Binomial distributions. We have then obtained the conditional coverage guarantee.

Assume now that the labels $Y_1, \ldots, Y_{n+1}$ are i.i.d.. Then $n_y = \sum_{i=1}^n \mathbf{1}\{Y_i = y\}$ is independent of $Y_{n+1}$ and follows a Binomial distribution $\mathcal{B}(n, q(y))$ for $q(y) := \mathbb{P}[Y_1 = y]$. Then:

$$\mathbb{E}\left[\frac{1}{n_{Y_{n+1}+1}}\right] = \mathbb{E}\left[\sum_{y=1}^K \frac{q(y)}{n_y + 1}\right] \leq \mathbb{E}\left[\sum_{y=1}^K \frac{q(y)}{q(y)(n+1)}\right] = \frac{K}{n+1} ,$$

where we have used Lemma F.3 for the inequality. That gives the marginal coverage after taking the expectation relatively to $Y_{n+1}$ into (21). $\qquad\square$

## B Failure of the naive approaches

We discuss in this section the challenges that naive approaches may face when dealing with the case of aggregated predictions.

### B.1 Synthetic dataset

We first consider two naive methods that we view as natural. We illustrate them in the case of mean aggregation and evaluate their performance on the synthetic dataset presented in Section 6.1.

**First method: directly using the calibration set.** A first naive idea is to construct the prediction set using the original scores and to use it directly with the aggregating scores, *i.e.,* when the aggregation

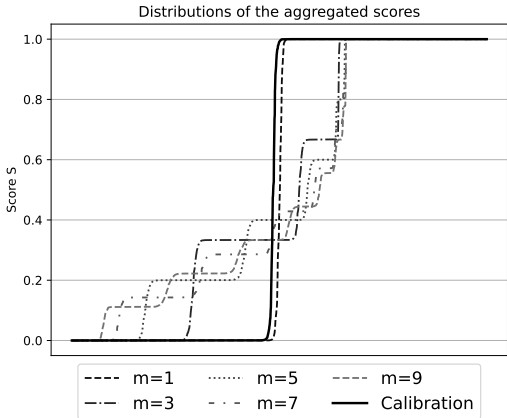

Figure 4: Distribution of the averaged softmax scores for synthetic data. 5000 averaged scores are computed and sorted for $m \in \{1, 3, 5, 7, 9\}$.

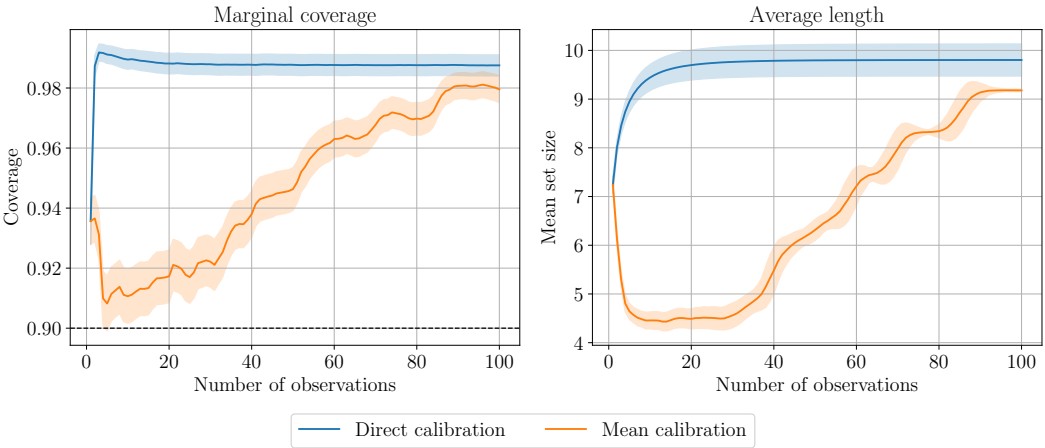

Figure 5: Empirical coverage and average lengths for class-conditional approach using two naive calibration sets for synthetic data. The calibration set is of size 5000, the coverage and the length are evaluated on 300 multi-inputs repeated 1000 times. The confidence region is $\pm$ (empirical std)$/\sqrt{5000}$

is the mean, we are considering the set $\left\{ y : S_m^{agg}(y) := m^{-1} \sum_{j=1}^{m} S_{n+j}(y) \leq S_{(\lceil (n_y+1)(1-\alpha) \rceil)}^{y} \right\}$. Formally, this set lacks any guarantee since the new (aggregated) score $S_m^{agg}(Y_{n+1})$ follows a different distribution than the calibration scores. In Figure 4, we compare the distribution of this averaged scores for $m = 1$ to 9 observations to the distribution of calibration scores. A clear change in distribution can be observed even for small value of $m$. In practice, this method produces conservative prediction sets, whose size increases with the number of repetitions, as it can be seen in Figure 5.

**Second method: aggregating the calibration scores.** A second idea to obtain a calibration set exchangeable with the new score is, for a given number of observations $m$, to aggregate the calibration scores by class, *i.e.,* by averaging them in groups of $m$ points, in order to mimic the distribution of aggregated scores. These new aggregated calibration scores will be exchangeable under Assumption 3.3 with the new aggregated score. However, by doing so, the calibration set is reduced to approximately $n/m$ points (more precisely, $\sum_{y=1}^{K} \lfloor n_y/m \rfloor$). Recall that the number of points per class must be at least $\alpha^{-1}$ in order to obtain informative class-conditionally valid prediction sets, which no longer holds when $m$ is large. As shown in Figure 5, this method yields valid prediction sets whose size decreases for small values of $m$, but increases again when the number of observations becomes too large. This strategy may be sufficient in cases involving a limited number of inputs, or for classes with a sufficient number of points, *i.e.,* when $m \ll n_y$. Note however that the methods based on

p-values aggregation outperform this strategy, with an average size quickly below $4$ elements (see Figure 2).

*Remark* B.1. Figure 4 also highlights the occurrence of tied scores produced by the softmax output of a neural network. When examining the calibration curve, constructed from one minus the softmax outputs corresponding to the true classes, we observe that many of these values are either close or equal to $0$ or $1$ (plateaus at the left or the right). It indicates that the classifier often returns a Dirac, sometimes in the true class (scores of $0$) but also often in a wrong class (scores of $1$). This leads to score ties, therefore caused by a combination of classifier overconfidence and numerical approximation.

## B.2 LifeClef dataset

Our approach assumes that, at the time of calibration, only individual data points are observed, rather than multi-inputs. If multi-inputs are available at calibration time, an alternative to our methods would be to apply a class-conditional conformal prediction approach to the multi-inputs. In this setting, each data point corresponds to a multi-input, and each calibration score is computed as the aggregated score of that multi-input. However, as with the "second method" presented above for the synthetic dataset, such an approach requires a sufficient number of multi-inputs per class, specifically at least $\alpha^{-1}$. In the case of the LifeCLEF 2015 dataset, these conditions are not satisfied, and consequently, this method yields prediction sets that include nearly all classes. In comparison, the p-value aggregation methods does not suffer this issue as it is constructed from the calibration scores taken individually. To overcome this issue, it could be worthwhile to explore class-conditional conformal prediction methods specifically adapted to long-tailed distributions, such as Ding et al. (2023, 2025) but this remains out of the scope of this work. Nevertheless, we emphasize that if a sufficient number of multi-inputs is available, we can expect this method to behave better.

This is illustrated in Table 2, which reports the results of this naive method applied to the LifeCLEF dataset constructed as described in Section 6.2 in average over all sizes and for different sizes of multi-inputs. The aggregation strategy remains a simple average of the softmax outputs. However, as previously discussed, the large prediction set sizes are due primarily to the limited amount of available data, rather than to the quality of the aggregation itself.

| $m$ | 1 | 3 | 5 | 7 | 9 | In average |
|---|---|---|---|---|---|---|
| Coverage (%) | $88.2 \pm 0.5$ | $96.3 \pm 0.6$ | $97.5 \pm 0.7$ | $95.5 \pm 1.5$ | $99.0 \pm 1.0$ | $91.9 \pm 0.3$ |
| Length | $623 \pm 1$ | $665 \pm 1$ | $673 \pm 1$ | $675 \pm 1$ | $677 \pm 1$ | $646 \pm 1$ |

Table 2: Evaluation of class-conditional conformal prediction using multi-inputs structure conditional of different size $m$ of multi inputs and marginally over the different sizes. Recall that $K = 688$.

## C Majority voting.

We present in this section the majority voting strategies proposed by Gasparin and Ramdas (2024) for aggregating conformal sets and the guarantee we get in our setting. The purpose of these methods is to aggregate marginally valid prediction sets $\mathcal{C}_1, \ldots, \mathcal{C}_m$ coming from some blackbox method. We have compared to two of their methods that we consider of interest for our application. The *majority vote set* $\mathcal{C}^M$ contains the classes selected by at least one half of the sets. Formally:

$$\mathcal{C}^M = \left\{ y : \frac{1}{m} \sum_{j=1}^m \mathbf{1}\{y \in \mathcal{C}_j\} \geq 1/2 \right\}. \tag{22}$$

If the sets $\mathcal{C}_j$ have a coverage of $1 - \alpha$, this set has a marginal coverage of at least $1 - 2\alpha$. To be able to compare this method to our sets of class-conditional coverage $1 - \alpha$, we apply it to the sets $\mathcal{C}_j = \mathcal{C}^{cd}_{\alpha/2}(X_{n+j})$ of class-conditional coverage $1 - \alpha/2$. Then, the resulting majority vote set has a class conditional coverage of at least $1 - \alpha$, *i.e.,* for all $y \in [\![K]\!]$:

$$\mathbb{P}\big[y \in \mathcal{C}^M | Y_{n+1} = y\big] = \mathbb{P}\left[\frac{1}{m} \sum_{i=1}^m \mathbf{1}_{y \in \mathcal{C}^{cd}_{\alpha/2}(X_{n+i})} \geq \frac{1}{2} \,\Big|\, Y_{n+1} = y\right] \geq 1 - \alpha.$$

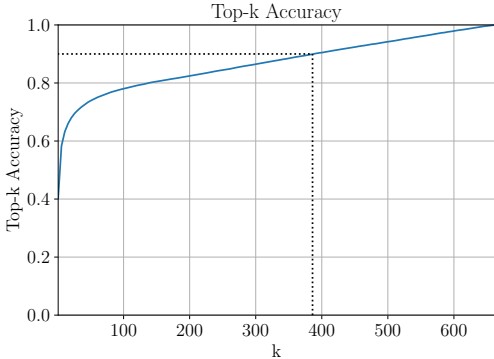

Figure 6: Top-k accuracy of the model for the LifeClef dataset. The dot line indicates the top-k accuracy of $0.9$: at least $k = 380$ classes are needed to reach this level.

This method can be improved when the sets are exchangeable (Gasparin and Ramdas, 2024): in that situation the intersection $\mathcal{C}^{Me}$ of the majority vote sets with increasingly more "voters" defined by:

$$\mathcal{C}^{Me} = \left\{ y : \forall k \in [\![m]\!] : \frac{1}{k} \sum_{j=1}^{k} \mathbf{1}\left\{ y \in \mathcal{C}^{cd}_{\alpha/2}(X_{n+i}) \right\} \geq 1/2 \right\} \tag{23}$$

ensures a class conditional coverage of at least $1 - \alpha$. In fact, Assumption 3.3 guarantees that the sets $\mathcal{C}^{cd}_{\alpha/2}(X_{n+i})$ for $i \in [\![m]\!]$ are exchangeable given $Y_{n+1}$.

## D   Further experiments and technical details for Section 6.2

### D.1   Technical details

The classifier considered is ResNet50, pre-trained on Pl@ntNet-300K Garcin et al. (2021) (see code and models at https://github.com/plantnet/PlantNet-300K/), for which we have fine tuned the last layer on our training set. The system we used for this training phase is equipped with a 4x Intel Xeon Gold 6142 (64 cores/128 threads total @ 2.6-3.7 GHz, 88MB L3 cache) while the GPUs are 2x NVIDIA A10 (24GB VRAM each) and 2x NVIDIA RTX 2080 Ti (11GB VRAM each), for a total of 70GB GPU VRAM. Obtaining the softmax scores required about 10 hours on our GPU resources to fine tune this pre-trained model. The conformal method runs on a standard machine. The codes of the experiments are provided in the supplementary materials.

Figure 6 presents the top-k accuracy of the model for different values of $k$. To achieve an accuracy of $0.9$, one needs to consider at least approximately $k = 380$ classes. This information can be compared with the average sizes of the conformal prediction sets shown in Figure 3. These sets tend to be significantly smaller for the methods we introduce, when a few additional observations are available. Even for $m = 1$ with a large temperature parameter (Temp=20), p-value-based methods produce smaller prediction sets, whose size is less than 100. This is likely due to the randomization involved in the p-value computation.

### D.2   Additional experiments

In Figure 7, we report the marginal coverage of the prediction sets, as well as the average class-conditional coverage. Due to the small number of observations available for some classes (*e.g.,* only 14 test points in the smallest class, which yields at most one multi-input sample of size $m = 10$ for this class), evaluating the conditional coverage per class becomes infeasible. In this case, the average class-conditional coverage serves as a proxy for the true conditional coverage. It is also worth noting that it closely aligns with the marginal coverage, which is expected given the limited number of test points.

In Figure 8, we present the results (coverage and size) of our methods applied using the APS score (Romano et al., 2020) instead of the softmax score used in the rest of our experiments. Let

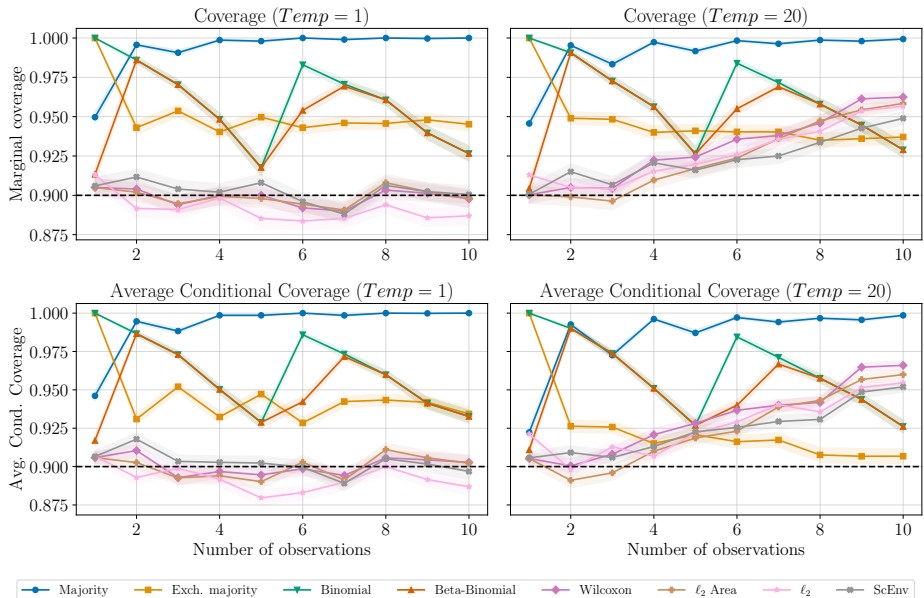

Figure 7: Coverage of the methods for the LifeClef dataset and different values of temperature.

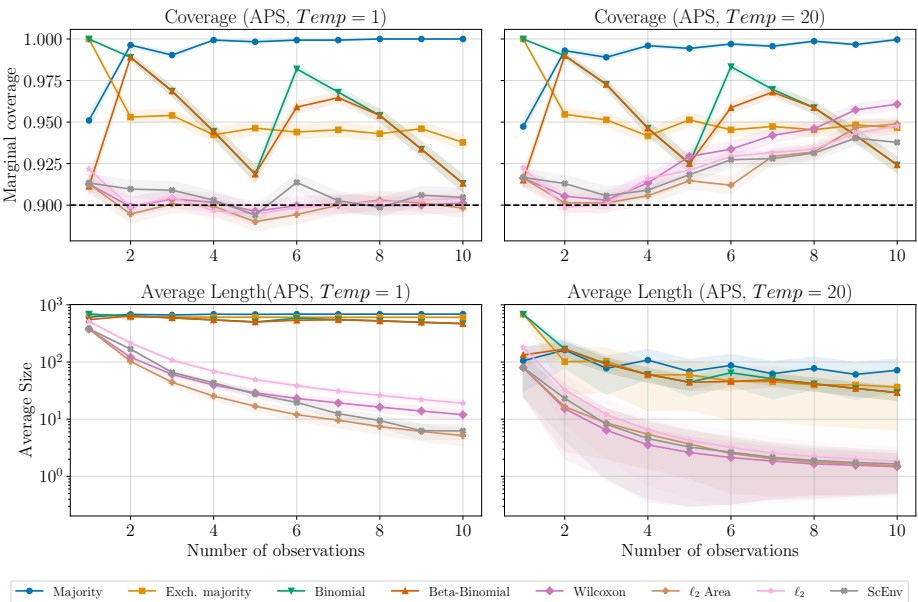

Figure 8: Coverage and size (in log scale) of the methods for the LifeClef dataset with APS score and different values of temperature.

$\widehat{\pi} : \mathcal{X} \to [0,1]^{|\mathcal{Y}|}$ be the classifer (softmax output of the neural network), it is defined as:

$$s_{\mathrm{APS}}(x, y) = \sum_{y' \in \mathcal{Y}} \widehat{\pi}(x)_{y'} \mathbf{1}_{\widehat{\pi}(x)_{y'} > \widehat{\pi}(x)_y} + U \widehat{\pi}(x)_y \,, \tag{24}$$

where $U \sim \mathcal{U}(0,1)$ is a uniform random variable. The results are similar to those obtained using the softmax score; the p-value aggregation methods offer again a substantial improvement in terms of size compared to the majority-based methods.

# E  Limitations of the assumption of exchangeability.

The methods proposed in this work rely on the assumption that the multi-input samples are exchangeable with data from the same class. In the context of Pl@ntNet, this means that multiple images of the same plant behave similarly to images of different plants from the same species. In the experiments presented above, this assumption is satisfied by construction of the dataset, as the multi-inputs are sampled within the same class.

The `LifeCLEF Plant Identification Task 2015` dataset also features a multi-input structure, derived from images jointly submitted by users. However, we observed that the exchangeability assumption does not hold for the multi-inputs in this dataset. The correlation between images within the same input is strong enough to break Assumption 3.3, resulting in prediction sets that are no longer valid.

Our p-value-based methods implicitly rely on the idea that each new observation brings additional information, thereby enabling more refined class selection. When images within the same input are too similar, the resulting p-values tend to be overly close, which can lead to the rejection of the correct class.

In contrast, majority vote-based methods are more robust to this effect and typically yield prediction sets with appropriate coverage, although often at the cost of larger size. The Binomial and Beta-Binomial methods still improve upon majority vote strategies (with a reduction in size of around 100 for $m = 10$), but, like the p-value aggregation approach, they no longer offer theoretical guarantees when the exchangeability assumption is violated.

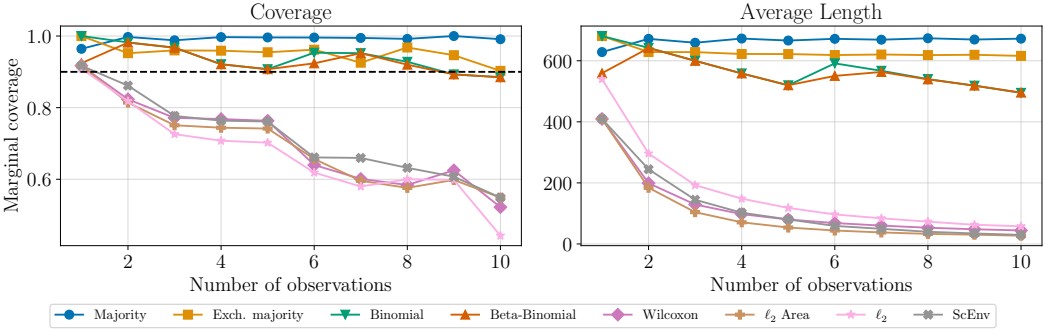

Figure 9: Empirical coverage and average lengths for the LifeClef dataset with multi-input breaking exchangeability.

The coverage and the average length of Figure 9 are evaluated for a number of multi-input observation from 4102 for $m = 1$ to 112 to $m = 10$. The temperature parameter is equal to 1. The loss of exchangeability causes a clear decrease in coverage of methods based on p-values aggregation.

# F  Probabilistic results

## F.1  Total variation distance between Binomial and Beta-Binomial distributions

The following Lemma gives a bound on the total variation distance between a Binomial and a Beta-Binomial distribution.

**Lemma F.1** (Teerapabolarn, 2008). *Let $m \in \mathbb{N}$, $\alpha, \beta \in \mathbb{R}_+$, if $p = \frac{\alpha}{\alpha+\beta}$ and $q = 1 - p$, then:*

$$d_{TV}\Big(\mathcal{B}(m,p), \mathtt{BetaBin}(m,\alpha,\beta)\Big) \leq \big(1 - p^{m+1} - q^{m+1}\big)\frac{m(m-1)}{(m+1)(1+\alpha+\beta)}.$$

Let us apply this bound to the conformal setting where `BetaBin` has parameters $m$, $k_\alpha$ and $n+1-k_\alpha$ with $k_\alpha = \lceil (n+1)(1-\alpha) \rceil$ and $\alpha \in (0,1)$.

**Corollary F.2.** *Let $k_\alpha = \lceil (n+1)(1-\alpha) \rceil$, then*

$$d_{TV}\left(\mathcal{B}\left(m, {}^{k_\alpha}/_{n+1}\right), \texttt{BetaBin}\left(m, k_\alpha, n+1-k_\alpha\right)\right) \leq \frac{(m-1)\min(1, m\alpha)}{n+2}.$$

*Proof.* It is directly deduced from Lemma F.1. Just lower bound $q$ by 0 and $p = \frac{k_\alpha}{n+1}$ by $1-\alpha$ to get that:

$$d_{TV}\left(\mathcal{B}\left(m, {}^{k_\alpha}/_{n+1}\right), \texttt{BetaBin}\left(m, k_\alpha, n+1-k_\alpha\right)\right) \leq \left(1-(1-\alpha)^{m+1}\right)\frac{m}{m+1}\frac{(m-1)}{n+2}$$

Then let us just remark that:

$$\frac{(m-1)m}{m+1}\left(1-(1-\alpha)^{m+1}\right) \leq \frac{(m-1)m}{m+1}\min(1, (m+1)\alpha) \leq (m-1)\min(1, \alpha m).$$

$\square$

## F.2 Technical lemma

**Lemma F.3.** *Let $N \sim \mathcal{B}(n, p)$, then:*

$$\mathbb{E}\left[\frac{1}{N+1}\right] = \frac{1}{(n+1)p}(1-(1-p)^{n+1}).$$

*Proof.* By direct computation:

$$\begin{aligned}
\mathbb{E}\left[\frac{1}{N+1}\right] &= \sum_{k=0}^{n} \frac{1}{k+1}\binom{n}{k}p^k(1-p)^{n-k} \\
&= \sum_{k=0}^{n} \binom{n+1}{k+1}\frac{1}{n+1}p^k(1-p)^{n-k} \\
&= \frac{1}{(n+1)p}\sum_{k=1}^{n+1}\binom{n+1}{k}p^k(1-p)^{n+1-k} \\
&= \frac{1}{(n+1)p}\left(1-(1-p)^{n+1}\right).
\end{aligned}$$

where the last equality holds because the sum is almost the full binomial expansion, except for the first term.

$\square$

