# OpenReview forum: "Class conditional conformal prediction for multiple inputs by p-value aggregation"
_NeurIPS.cc/2025/Conference — NeurIPS 2025 poster_

### Official Review · Reviewer_WJtv · 2025-07-01

**Clarity:** 3
**Significance:** 2
**Originality:** 2
**Rating:** 4
**Confidence:** 4

**Summary:**

This paper addresses class-conditional conformal prediction with multiple input covariates. Building upon the perspective of p-values and mathematical tools for their joint distribution, the paper proposes techniques to merge multiple prediction sets by aggregation methods including and beyond majority voting, which corresponds to different choices of non-conformity score functions, and reduces the size of the uncertainty set compared to the original majority voting. Under the critical assumption that the scores within the same class are exchangeable, class-conditional coverage is guaranteed. The technique is validated by experiments on both synthetic data and realistic data from citizen science, with simulated multiple inputs.

**Questions:**

1. Is it possible to empirically validate the technique with natural datasets containing multiple inputs, instead of synthetic construction of multiple inputs?

**Ethical Concerns:**

["NO or VERY MINOR ethics concerns only"]

**Final Justification:**

Considering the feedback from the authors, I maintain my vote for acceptance (marginal above the threshold).

The major contribution of the paper is to address conformal prediction with multi-view inputs.  The proposed framework unifies and generalizes aggregation methods for prediction sets through various choices of score functions taking p-values as input.

The limitation remains over the class-conditional exchangeability assumption. Thank the author for their clarification and reference to the appendix. In fact, the exchangeability assumption does not hold for the LifeCLEF Plant Identification Task 2015 dataset.

**Limitations:**

yes

**Quality:**

3

**Strengths And Weaknesses:**

### Strengths
1. The paper has unified aggregation methods for prediction sets as forming an envelope around their p-values. Subsequently, it is well-motivated from the observation that the envelope for majority voting is not tight enough. The paper generalizes aggregation methods through various choices of score functions taking p-values as input. The resulting framework reduces the size of the prediction set by design, and is also empirically validated.

### Weaknesses
1. The critical assumption that the scores within the same class are exchangeable (Assumption 3.3) weakens the applicability of the technique. In the context of citizen science, it implies an impractical assumption that images of the same plant are exchangeable with images of different plants from the same species. The paper also reports that the exchangeability assumption does not hold for the LifeCLEF Plant Identification Task 2015 dataset. Therefore, exchangeability is enforced by random grouping of images of the same class. Empirical results on natural datasets without synthetic transformation would be more convincing to test the robustness of the technique when the assumption is violated.

### Comments
1. The core mathematical tool for the exact distribution of multiple values is mostly based on Gazin et al, though the results have been extended to handle ties between scores, and directly extended to be conditioned on classes.

---

> ### Author Rebuttal · Authors · 2025-07-31
>
> We thank the reviewer for the feedback and positive review. As discussed below, we would like to point out that we have evaluated our method on natural dataset in Appendix E to observe the limitation of the exchangeability assumption.
>
> **Comments**
>
> *The core mathematical tool for the exact distribution of multiple values is mostly based on Gazin et al, though the results have been extended to handle ties between scores, and directly extended to be conditioned on classes.*
>
> We would nonetheless like to emphasize that, although our main result builds on that of Gazin et al. (2024), its application in the context of conformal prediction for multi-input settings is entirely novel. As the reviewer rightly pointed out, our theoretical contributions lie in the treatment of ties and the class-conditional use of the tool.
>
> **Questions:**
>
> *Is it possible to empirically validate the technique with natural datasets containing multiple inputs, instead of synthetic construction of multiple inputs?*
>
> We have evaluated our method on the LifeClef dataset without artificially sampling multi-inputs and observed that the exchangeability assumption is violated in this case, as detailed in Appendix E. As our method works well under this assumption, our focus is now more on relaxing the exchangeability assumption than on identifying datasets that satisfy it. We believe we have been transparent about this limitation and intend to address it in future work.
>
> We remain available during the discussion phase if anything remains unclear.

---

> > ### Comment · Reviewer_WJtv · 2025-08-09
> >
> > Considering the feedback from the authors, I maintain my vote for acceptance (marginal above the threshold).
> >
> > The major contribution of the paper is to address conformal prediction with multi-view inputs.  The proposed framework unifies and generalizes aggregation methods for prediction sets through various choices of score functions taking p-values as input.
> >
> > The limitation remains over the class-conditional exchangeability assumption. Thank the author for their clarification and reference to the appendix. In fact, the exchangeability assumption does not hold for the LifeCLEF Plant Identification Task 2015 dataset.

---

### Official Review · Reviewer_RZJ6 · 2025-07-01

**Clarity:** 3
**Significance:** 3
**Originality:** 4
**Rating:** 5
**Confidence:** 4

**Summary:**

This paper presents a novel framework for reducing uncertainty in class-conditional conformal prediction, specifically tailored to scenarios where multiple observations (e.g., images) of a single instance are available at test time. This setting is well-motivated by citizen science applications such as Pl@ntNet. The core idea is to perform class-conditional conformal prediction on each individual observation to generate p-values, and then aggregate these p-values to make a final prediction.

The authors first propose a refinement of majority voting by replacing the heuristic threshold with a quantile from the Beta-Binomial distribution. Second, they introduce a more powerful class of aggregation methods based on score functions defined directly on the vector of p-values. These methods—including scoring based on quantiles or the geometric area of the p-value trajectory—better capture joint information from multiple observations.

The authors validate their methods on synthetic data and the LifeCLEF 2015 plant identification dataset, demonstrating a significant reduction in prediction set size compared to baselines, while maintaining the desired class-conditional coverage guarantees.

**Questions:**

- **Line 49**: In your setting description, is *m* (the number of observations) considered fixed or a random variable? In practice, users may submit different numbers of images. Does the framework still hold if *m* varies per test instance, as long as calibration for the p-value scores (Algorithm 1) is done for that specific *m*? This should be clarified.

### Typos
- **Line 161**: “is” should be “are.”
- **Line 143**: “conditionnal” should be “conditional.”

**Ethical Concerns:**

["NO or VERY MINOR ethics concerns only"]

**Final Justification:**

The authors addressed the main concerns, notably by adding experiments with APS scores and committing to include confidence intervals in the plots. Their explanations regarding methodology and evaluation were satisfactory, and the remaining limitations do not significantly affect the overall contribution. Based on this, I maintain my recommendation to accept.

**Limitations:**

The authors are transparent about the main assumption underpinning their framework (Assumption 3.3), and they strengthen the paper by providing an empirical analysis in Appendix E where this assumption is violated.

A second, less discussed limitation is the potential computational cost of the p-value scoring methods (Section 5), which require separate Monte Carlo calibrations for each class. This may become prohibitive in settings with a very large number of classes (K), although the calibration is a one-time offline process.

**Quality:**

3

**Strengths And Weaknesses:**

### Strengths
- The paper addresses uncertainty quantification in a setting where multiple inputs are available at test time, a scenario where standard conformal prediction is not directly applicable (as discussed in Appendix B).

- The derivation of the joint distribution of ordered conditional p-values (Theorem 3.4) provides a solid foundation for Sections 4 and 5.

- The paper offers two methodological contributions: (1) a refined majority voting scheme using the Beta-Binomial distribution (Section 4), which is a direct and elegant improvement over existing heuristics; and (2) a general framework for scoring p-value vectors based on simulation (Section 5), which flexibly captures the structure of joint p-values.

- Experiments on both synthetic and real-world data show that the proposed methods, particularly the p-value scoring envelopes, produce significantly smaller prediction sets than baselines, while maintaining valid coverage.

### Weaknesses
- A major concern is the strong assumption of exchangeability among the multiple inputs (i.e., views) of a test instance. In real-world multi-view settings—such as images taken from different angles, lighting conditions, or crops—this assumption is often violated due to inherent correlations between views. For example, an image and its 90° rotation are not identically distributed and thus not exchangeable. The theoretical guarantees of the proposed framework rest on Assumption 3.3, which requires exchangeability between test-time inputs and the calibration data from the same class. To sidestep this issue, the experimental setup simulates multi-view data using independently drawn samples from the same class, thereby artificially restoring exchangeability. While this limitation is acknowledged, it raises significant doubts about whether the method can be reliably applied to realistic multi-view scenarios, particularly in cases where users submit similar or low-quality images.

- The evaluation primarily compares the proposed methods to majority voting—a simple heuristic that does not take into account the scale of p-values. For a fairer and more comprehensive comparison, it would be valuable to benchmark against more advanced aggregation techniques.

- Most plots comparing coverage and prediction set sizes (except for Figure 5) do not include any indication of statistical uncertainty (e.g., error bars or confidence intervals). This limits the reader's ability to assess the robustness and significance of the reported improvements.

- While the framework is general, the empirical validation is limited to basic softmax scores. Demonstrating the method’s effectiveness with stronger conformity scores, such as Adaptive Prediction Sets (APS) [1], would reinforce the claims—especially since APS has been shown to improve conditional coverage.

- In Figure 2, the refined voting methods (Beta-Binomial and Binomial) consistently exhibit over-coverage, providing coverage significantly above the nominal 90% level. A discussion of the causes and potential corrections for this behavior would strengthen the analysis in Section 4.

- Some methodological choices lack justification. For instance, the choice of quantile level $\lambda$ in the quantile envelope method (line 271) is not explained. Similarly, the decision to approximate the Beta-Binomial distribution with a Binomial (Proposition 4.2) is unclear, particularly given that Beta-Binomial quantiles are efficiently computable (e.g., via `scipy.stats.betabinom`).

- While the main results are strong, additional metrics could provide further insights. Reporting the median prediction set size (e.g., in the Appendix) would offer robustness to outliers. Furthermore, evaluating coverage conditional on the input features (e.g., worst-slab coverage, as reported in [1]) could show whether the aggregation methods improve this stricter form of conditional validity.

[1] Romano, Yaniv et al. "Classification with valid and adaptive coverage." NeurIPS 2020.

---

> ### Author Rebuttal · Authors · 2025-07-31
>
> We thank the reviewer for the positive feedback and detailed review. Below, we provide clarifications on the points that remain unclear and further discuss the weaknesses raised. We have revised the paper accordingly in light of the suggestions provided by the reviewer.
>
> **Weaknesses**
>
> *A major concern is the strong assumption of exchangeability among the multiple inputs (i.e., views) of a test instance. In real-world multi-view settings—such as images taken from different angles, lighting conditions, or crops—this assumption is often violated due to inherent correlations between views. For example, an image and its 90° rotation are not identically distributed and thus not exchangeable. The theoretical guarantees of the proposed framework rest on Assumption 3.3, which requires exchangeability between test-time inputs and the calibration data from the same class. To sidestep this issue, the experimental setup simulates multi-view data using independently drawn samples from the same class, thereby artificially restoring exchangeability. While this limitation is acknowledged, it raises significant doubts about whether the method can be reliably applied to realistic multi-view scenarios, particularly in cases where users submit similar or low-quality images.*
>
> We are aware of this limitation and have made an effort to be transparent about it. We are really interested in addressing it, but believe that this challenge warrants a dedicated study in future work.
>
> *The evaluation primarily compares the proposed methods to majority voting—a simple heuristic that does not take into account the scale of p-values. For a fairer and more comprehensive comparison, it would be valuable to benchmark against more advanced aggregation techniques.*
>
> We chose to compare our method to majority voting, as it has been used in conformal prediction for aggregating prediction sets. We considered that standard p-value aggregation methods, such as Bonferroni or Simes, would yield similar results based on the observations in Figure 1. However, we are open to testing these methods and welcome any suggestions from the reviewer regarding alternative p-value aggregation strategies.
>
> *Most plots comparing coverage and prediction set sizes (except for Figure 5) do not include any indication of statistical uncertainty (e.g., error bars or confidence intervals). This limits the reader's ability to assess the robustness and significance of the reported improvements.*
>
> Confidence intervals have so far been computed only on synthetic data. For real datasets, we plan to include them in the camera-ready version by repeatedly shuffling the calibration and test sets.
>
> *While the framework is general, the empirical validation is limited to basic softmax scores. Demonstrating the method’s effectiveness with stronger conformity scores, such as Adaptive Prediction Sets (APS) [1], would reinforce the claims—especially since APS has been shown to improve conditional coverage.*
>
> We previously tested the use of the APS score and obtained results similar to those with softmax scores. This score is also implemented in the provided code (see supplementary material). We will include these experiments in the camera-ready version.
>
> *In Figure 2, the refined voting methods (Beta-Binomial and Binomial) consistently exhibit over-coverage, providing coverage significantly above the nominal 90% level. A discussion of the causes and potential corrections for this behavior would strengthen the analysis in Section 4.*
>
> This phenomenon arises from the discrete nature of the statistic that counts the number of sets containing each label. As a result, an exact quantile at level $1- \alpha$ may not exist, leading to coverage that exceeds the threshold. In the case of the Beta-Binomial quantile, which corresponds to the exact distribution of the statistic, this issue can be addressed through additional randomization to achieve exact coverage at level $1- \alpha$. For the Binomial quantile, however, the situation is less straightforward, as it is an approximation of the true distribution and does not achieve a marginal coverage for large $m$.
>
> *Some methodological choices lack justification. For instance, the choice of quantile level in the quantile envelope method (line 271) is not explained. Similarly, the decision to approximate the Beta-Binomial distribution with a Binomial (Proposition 4.2) is unclear, particularly given that Beta-Binomial quantiles are efficiently computable (e.g., via scipy.stats.betabinom).*
>
> These choices stem from different aggregation methods proposed in the literature. As briefly discussed in Remark 4.3, the use of a Binomial quantile is connected to the aggregation method of Gasparin and Ramdas (2024) in the case of independent sets. We show that this choice also remains valid when the sets are correlated through the calibration dataset.
> The quantile envelope, sometimes referred to as a template, is commonly used in multiple testing procedures (see, e.g., Blanchard et al., 2024; Li et al., 2024) or in conformal prediction for ranking tasks (Fermanian et al., 2025), see discussion in Section 2.3. This envelope is intended to capture the behavior of each marginal by choosing for each coordinate the quantile of the marginal distribution at a common level $\lambda_\alpha$. We link this envelope to the score function $v_Q$ and this level $\lambda_\alpha$ is seen as a threshold of the score chosen using a conformal step.
>
> *While the main results are strong, additional metrics could provide further insights. Reporting the median prediction set size (e.g., in the Appendix) would offer robustness to outliers. Furthermore, evaluating coverage conditional on the input features (e.g., worst-slab coverage, as reported in [1]) could show whether the aggregation methods improve this stricter form of conditional validity.*
>
> Thank you for the recommendation. We have computed the median of lengths for the different methods and number of multi-inputs $m$, and observed results very similar to the behavior of the means. We will include this metric in the Appendix.
>
> **Questions**
>
> *Line 49: In your setting description, is m (the number of observations) considered fixed or a random variable? In practice, users may submit different numbers of images. Does the framework still hold if m varies per test instance, as long as calibration for the p-value scores (Algorithm 1) is done for that specific m? This should be clarified.*
>
> Thank you for the remark. We will clarify this point in the revised version. We would like to point out that our motivated application is a case where $m$ effectively varies at test time. In Pl@ntNet, users may submit a different number of images. However, Pl@ntNet has access to only a (relatively) small number of trustworthy labeled multi-input and relies only on lonely labeled images of plants, which motivates the setting considered. In our method, the thresholds (exact quantiles of simulated ones) of all our methods depend on $m$ and are then recomputed (or loaded) depending on the size of the new multi-input.
>
> Theoretically, we have supposed that $m$ is fixed. However, as long as Assumption 3.3 holds conditionally on $m$, all the theoretical results remain valid, can be stated conditionally on $m$, and then are marginally valid with respect to the randomness of $m$.
>
> **Typos**
>
> Thanks for the typos. We have updated the paper accordingly.
>
> **Limitations:**
>
> *A second, less discussed limitation is the potential computational cost of the p-value scoring methods (Section 5), which require separate Monte Carlo calibrations for each class. This may become prohibitive in settings with a very large number of classes (K), although the calibration is a one-time offline process.*
>
> The quantile of the score $v$ must indeed be computed for each class. However, we do not consider this computationally limiting for several reasons:
> - The p-values simulation relies solely on a draw without replacement (Algorithm 1), which is both easily parallelizable and computationally inexpensive ($O(n_y + m)$ for class $y$) where $n_y$ is the number of points of calibration of class $y$..
> - As for an application, the number of points by class $n_y$ in the calibration is fixed, although this is not formally valid; these quantiles can be precomputed and stored for different values of $m$. Since the distribution depends only on $n_y$, the number of points per class, and $m$, the number of multi-inputs, these quantiles can even be reused for other applications.
> - For experiments on PlantClef, the number of classes is around 700, and the simulation can be run on a laptop. Moreover, in practice, the number of repetitions $m$ required is typically small.
>
> We remain available during the discussion phase if the reviewer wishes to further discuss these points.
>
> [1] Romano, Yaniv et al. "Classification with valid and adaptive coverage." NeurIPS 2020.

---

> > ### Comment · Reviewer_RZJ6 · 2025-08-04
> >
> > Thank you for the detailed rebuttal. I am satisfied with the proposed explanations and revisions, including the addition of experiments with APS scores and confidence intervals in the plots for the final version. I maintain my recommendation to accept.

---

### Official Review · Reviewer_Joh5 · 2025-07-02

**Clarity:** 4
**Significance:** 4
**Originality:** 4
**Rating:** 5
**Confidence:** 5

**Summary:**

This paper proposes a framework to integrate multi-input information into conformal prediction to further improve the predictive interval.

**Questions:**

Your results for the LifeCLEF dataset show a significant difference in performance based on the temperature scaling of the softmax outputs. This suggests a strong interplay between the quality of the base classifier's scores and the effectiveness of your aggregation. Could you expand on this? For example, would applying a more sophisticated calibration method (like Ensemble Temperature Scaling) to the base classifier's scores before computing p-values further improve the reduction in set size?

**Ethical Concerns:**

["NO or VERY MINOR ethics concerns only"]

**Final Justification:**

My score remains unchanged. Overall this is a solid paper.

**Limitations:**

Yes.

**Paper Formatting Concerns:**

N/A.

**Quality:**

4

**Strengths And Weaknesses:**

Strength: original; I also enjoyed reading this paper.

Weakness. If I understand correctly,Section 5 p-value score aggregation requires a Monte Carlo simulation to calibrate the quantiles of the score v for every class? For datasets with thousands of classes, this could be computationally prohibitive. The paper does not discuss the computational complexity or potential optimizations for this critical step.

---

> ### Author Rebuttal · Authors · 2025-07-31
>
> We thank the reviewer for the positive feedback and provide answer to your questions below.
>
> **Weakness.** *If I understand correctly, Section 5 p-value score aggregation requires a Monte Carlo simulation to calibrate the quantiles of the score v for every class? For datasets with thousands of classes, this could be computationally prohibitive. The paper does not discuss the computational complexity or potential optimizations for this critical step.*
>
> The quantile of the score $v$ must indeed be computed for each class. However, we do not consider this computationally limiting for several reasons:
> - The p-values simulation relies solely on a draw without replacement (Algorithm 1), which is both easily parallelizable and computationally inexpensive ($O(n_y + m)$ for class $y$) where $n_y$ is the number of points of calibration of class $y$..
> - As for an application, the number of points by class $n_y$ in the calibration is fixed, although this is not formally valid; these quantiles can be precomputed and stored for different values of $m$. Since the distribution depends only on $n_y$, the number of points per class, and $m$, the number of multi-inputs, these quantiles can even be reused for other applications.
> - For experiments on PlantClef, the number of classes is around 700, and the simulation can be run on a laptop. Moreover, in practice, the number of repetitions $m$ required is typically small.
>
> *Your results for the LifeCLEF dataset show a significant difference in performance based on the temperature scaling of the softmax outputs. This suggests a strong interplay between the quality of the base classifier's scores and the effectiveness of your aggregation. Could you expand on this? For example, would applying a more sophisticated calibration method (like Ensemble Temperature Scaling) to the base classifier's scores before computing p-values further improve the reduction in set size?*
>
> In general, a well-performing classifier is expected to produce prediction sets of small size. For a perfect classifier, the prediction set would contain only the true class. In our case, temperature scaling does not improve top-$k$ performance, since the transformation preserves the ordering of the scores. However, it appears that the smoothing effect of temperature scaling helps break ties between class scores caused by numerical approximations (by $0$ or $1$) in the softmax function . Such ties can negatively impact conformal prediction methods by producing overly large prediction sets.
>
> In scenarios where the classifier is overly confident yet incorrect, the softmax scores may degenerate into binary values ($0$ or $1$), which leads to excessively large prediction sets (see, for instance, the curve “calibration” in Figure 4 representing the values of the scores of the synthetic data.).
>
> To summarize:
> - In our setting, the main effect of temperature scaling seems to break ties between scores caused by numerical approximation.
> - More generally, any calibration method that improves the classifier’s performance is likely to be beneficial for the construction of prediction sets.
>
> We remain available during the discussion phase if the reviewer wishes to further discuss these points.

---

> > ### Comment · Reviewer_Joh5 · 2025-08-06
> >
> > Thank you for the response. I will maintain my score.

---

### Official Review · Reviewer_4nuN · 2025-07-03

**Clarity:** 3
**Significance:** 3
**Originality:** 3
**Rating:** 4
**Confidence:** 4

**Summary:**

This paper develops a conformal prediction framework for multiple observations of the same instance, motivated by citizen science applications like plant identification. The authors aggregate conformal p-values from individual observations rather than predictions directly, preserving exchangeability assumptions. Key contributions include deriving the exact joint distribution of class-conditional conformal p-values and proposing two aggregation strategies: refined majority voting using Beta-Binomial quantiles and p-value aggregation methods with various score functions.

**Questions:**

**Missing Baseline:** The experiments lack comparison with multi-view models combined with standard conformal prediction. This baseline would involve end-to-end multi-view architectures applying standard conformal prediction to their outputs. Without this comparison, the relative advantages of p-value aggregation versus direct multi-view modeling remain unclear. Could you provide guidance on when each approach is preferable?

**Exchangeability:** How sensitive are coverage guarantees to violations of Assumption 3.3? What diagnostic tools could practitioners use to assess this assumption?

**Score Function Selection:** What guidance exists for choosing between quantile, area-based, and ℓq-envelope score functions under different conditions?

**Ethical Concerns:**

["NO or VERY MINOR ethics concerns only"]

**Final Justification:**

The authors' rebuttal partially addresses my concerns, particularly regarding their experimental setting where multi-inputs are only available at test time, which is reasonable given the Pl@ntNet application context. However, the missing baseline comparison with multi-view models combined with standard conformal prediction remains a significant limitation that weakens the evaluation. While the authors acknowledge this gap and express willingness to include such comparisons, the current work lacks comprehensive baselines to demonstrate when p-value aggregation is preferable over direct multi-view modeling. Despite these limitations, the work addresses a practically important problem with novel theoretical insights, and the authors show a willingness to address the evaluation gaps. Given the theoretical rigor and potential impact, I raise my score.

**Limitations:**

The authors adequately address several important limitations, particularly the exchangeability assumption and its empirical violations. The computational requirements discussion is appropriate. However, more comprehensive baseline comparisons and specific practical implementation guidance would strengthen the work.

**Quality:**

3

**Strengths And Weaknesses:**

**Quality:** The theoretical development is rigorous with proper handling of exchangeability assumptions and tied scores. However, the experimental evaluation lacks a critical baseline comparing against multi-view models with standard conformal prediction.

**Clarity:** Well-written with clear mathematical exposition, though some technical details could be better integrated into the main text.

**Significance:** Addresses a practically important problem with meaningful theoretical extensions to conformal prediction. Empirical results show substantial improvements within the tested framework.

**Originality:** The p-value aggregation approach represents a novel contribution with original theoretical analysis of joint p-value distributions.

---

> ### Author Rebuttal · Authors · 2025-07-31
>
> We thank the reviewer for their feedback and understand their concern. We would like to point out that the comparison with standard conformal prediction using aggregated scores is discussed in Appendix B, and the limitations of the exchangeability assumption in Appendix E. Thanks to the reviewer’s feedback, we will further expand these discussions.
>
> **Missing Baseline:** *The experiments lack comparison with multi-view models combined with standard conformal prediction. This baseline would involve end-to-end multi-view architectures applying standard conformal prediction to their outputs. Without this comparison, the relative advantages of p-value aggregation versus direct multi-view modeling remain unclear. Could you provide guidance on when each approach is preferable?*
>
> We would like to draw the reviewer's attention to the fact that in our setting, multi-inputs are only present at test time. During both training and calibration, the available data consists solely of individual images. This design choice is motivated by the application context of Pl@ntNet, where the verified data primarily consists of single images. We refer the reviewer to the Pl@ntNet website, where it can be observed that the number of available images ($3.4 \times 10^7$) is very close to the number of multi-input observations ($2.8 \times 10^7$) so with an average number of multi-inputs of $1.2$. Nevertheless, our objective was thus to develop a method capable of providing meaningful prediction sets for multi-input sizes that may not have been encountered during training.
> Of course, this does not prevent us from comparing our approach to a standard prediction method that has access to multi-inputs. The advantage of our aggregation methods lies in the number of data points required for calibration. As a reminder, class-conditional conformal prediction methods typically require at least $\simeq \alpha^{-1}$ samples per class to yield non-trivial results. If the user has access to this number of multi-view samples per class, standard conformal prediction methods are expected to perform adequately.
>
> However, in practice, we have very few available calibration samples for certain classes, possibly even fewer than $m$, the size of the multi-input. This situation typically arises in long-tail datasets, of which the Pl@ntNet dataset is a representative example.
> By comparing prediction errors individually and aggregated, our methods are capable of handling such cases in a non-naive manner. For instance, our methods perform well even when the multi-input size exceeds the total number of calibration points available for a given class.
>
> We discussed this advantage in Appendix B (second naive approach: mean calibration), and indeed observed that standard conformal prediction methods tend to fail when $m$ becomes too large relative to the available calibration data. This comparison is carried out using a naive aggregation method (averaging the predictions), but the observation extends to more sophisticated approaches, as the limitation stems more from the number of available calibration samples than from the specific aggregation strategy used for the multi-view inputs.
>
> We are, however, open to comparing our method with standard CP applied to more sophisticated aggregation strategies. We would be particularly interested in any references to aggregation methods, especially those that are adaptive in $m$.
>
> **Exchangeability:** *How sensitive are coverage guarantees to violations of Assumption 3.3? What diagnostic tools could practitioners use to assess this assumption?*
>
> This assumption is central to our methodology and is further examined in Appendix E, where we apply our approach to data that do not satisfy it. We observe a loss in coverage of p-values based methods.
> If a few labeled multi-input samples are available, one may attempt to assess the exchangeability of the data, for example using Mann–Whitney–Wilcoxon tests.
> While relaxing this assumption is essential for broader applicability, we view the present work as a meaningful and necessary first step to this problem.
>
> **Score Function Selection:** *What guidance exists for choosing between quantile, area-based, and $\ell_q$-envelope score functions under different conditions?*
>
> These score functions yield comparable results across the different settings in which we have used them. We do not have strong recommendations in favor of one over the others. The quantile score has sometimes performed slightly better than the others in some of our experiments. However, we did not find these results to be significant relative to the simplicity of the area-based and $\ell_q$ score functions.
>
> We hope we have addressed your concerns and are happy to discuss further if not.

---

> > ### Comment · Reviewer_4nuN · 2025-08-06
> >
> > Thank you for your detailed rebuttal. I appreciate your clarification regarding the experimental setting where multi-inputs are only available at test time motivated by the Pl@ntNet. While I understand the rationale behind your current experimental design, I believe that including the missing baseline comparison would significantly strengthen the paper's contribution. Comparing your p-value aggregation approach against end-to-end multi-view models with standard conformal prediction would help readers understand the relative trade-offs and when each approach is preferable. Even though your setting is constrained by single-image training data, it would still be feasible to implement multi-view baselines at test time using existing pre-trained models or fine-tuning on aggregated features. If you could provide experimental results comparing against multi-view baselines, I would be willing to raise my score, as this would address my primary concern.

---

> > > ### Author Response · Authors · 2025-08-06
> > > **Answer to comment of reviewer 4nuN**
> > >
> > > Thank you for your response. We understand the reviewer’s concerns. We would appreciate it if the reviewer could point us to any relevant papers and available code implementations for classifier architectures in multi-input settings that they consider to be standard baselines. We will gladly include comparisons with these methods in the revised version of the paper and will also aim to provide additional experiments before the end of the discussion period.

---

### Decision · Program_Chairs · 2025-09-17

**Decision:**

Accept (poster)

**Comment:**

This paper provides a framework for reducing uncertainty in class-conditional conformal prediction, tailored to scenarios with multiple observations of a single instance. Contributions include deriving the exact joint distribution of class-conditional conformal p-values and proposing refined majority voting using Beta-Binomial quantiles and p-value aggregation methods with various score functions as aggregation strategies. The authors conduct experiments evaluating their approach on synthetic data and the LifeCLEF 2015 plant identification dataset and observe how prediction set size is affected by their techniques.

The reviewers appreciated the rigorous theoretical development and how the framework addresses a practically important problem, and the authors’ rebuttals clarified some questions reviewers had. In revising, I recommend the authors consider adding experimental comparisons to more sophisticated aggregation approaches than majority vote, as well as application of standard conformal prediction to a multiview system as described by reviewer 4nuN to demonstrate when the aggregation of p-values that the framework uses has an advantage.